# Neurocomputational mechanisms of biased impression formation in lonely individuals

Gabriele Bellucci [1,2✉] & Soyoung Q. Park[2,3,4,5]

Social impressions are fundamental in our daily interactions with other people but forming accurate impressions of our social partners can be biased to different extents. Loneliness has previously been suggested to induce biases that hinder the formation of accurate impressions of others for successful social bonding. Here, we demonstrated that despite counterfactual evidence, negative first impressions bias information weighting, leading to less favorable trustworthiness beliefs. Lonely individuals did not only have more negative expectations of others' social behavior, but they also manifested a stronger weighting bias. Reduced orbitofrontal cortex (OFC) activity was associated with a stronger weighting bias in lonelier individuals and mediated the relationship between loneliness and this weighting bias. Importantly, stronger coupling between OFC and temporoparietal junction compensated for such effects, promoting more positive trustworthiness beliefs especially in lonelier individuals. These findings bear potential for future basic and clinical investigations on social cognition and the development of clinical symptoms linked to loneliness.

[1] Department of Psychology, Royal Holloway, University of London, Egham TW20 0EX, UK. [2] Department of Psychology I, University of Lübeck, Lübeck, Germany. [3] Department of Decision Neuroscience and Nutrition, German Institute of Human Nutrition (DIfE), Potsdam-Rehbruecke, Nuthetal, Germany. [4] Charité-Universitätsmedizin Berlin, Corporate member of Freie Universität Berlin, Humboldt-Universität zu Berlin, and Berlin Institute of Health, Neuroscience Research Center, 10117 Berlin, Germany. [5] Deutsches Zentrum für Diabetes, 85764 Neuherberg, Germany. ✉email: gabriele.a.bellucci@gmail.com

First impressions and, in particular, impressions of other people's trustworthiness are central to social beliefs underlying many social behaviors[1,2]. Honesty is an important determinant of trustworthiness impressions and previous studies have shown that honesty-based impressions of a partner's trustworthiness generalize to different social contexts[3]. The formation of accurate impressions, however, can be biased by different factors during learning.

In particular, feelings of loneliness have been hypothesized to negatively impact the formation of positive impressions of social partners[4]. A model of how loneliness induces cognitive biases in social evaluations hypothesizes that feelings of loneliness are associated with hypervigilance for negative social cues, jeopardizing one's social abilities[5] and contributing to the development of depression[6,7]. Previous evidence indicates attentional, evaluative and memory biases that lead lonely individuals to pay more attention to threatening events, give more evaluative weight to negative interactions and be more likely to remember negative interactions[8–10]. Such cognitive biases have been proposed to be central to social withdrawal in lonely individuals, as they foster unfavorable expectations of others that induce lower levels of trust[11], more negative trust beliefs[12], and avoidance of social relationships[13].

Recently, it has been proposed that such biased social impressions impair social behaviors and feedback learning in social interactions, leading lonely individuals to form more negative expectations of social partners[14]. However, empirical evidence on whether and how loneliness affects impression formation and impacts learning dynamics underlying social beliefs in repeated social interactions is still lacking. In particular, first impressions are influenced by an array of variables, like temporal ordering that leads to the well-known primacy effect[15–19]. In social contexts, the primacy effect describes the phenomenon according to which initial information learnt at the beginning of an interaction with a new social partner biases the overall valence of the resulting impression of that person. Such effect is due to a weighting pattern biased toward the integration of subsequent information consistent with the valence of the very first pieces of information encountered[20]. Hence, initial negative information about a person endorses the formation of an overall more negative impression via biased updating of subsequent behavioral signals consistent with that initial negative information[21,22]. Interestingly, such primacy effect might be further strengthened by a person's initial expectations of social partners prior to any interaction[23,24].

This behavioral evidence on the effects of early behavioral signals and initial expectations on impression formation and learning in social interactions, and the hypothesized associations between loneliness and biased social expectations suggest that heightened feelings of loneliness might enhance the primacy effect induced by the initial information encountered about a social partner, especially when that information is negatively valenced. Thus, when learning about a partner's trustworthiness based on their honest behavior, feelings of loneliness might negatively bias the updating of social information toward negative instances, thereby leading to more negative impressions of the partner's trustworthiness, especially for those partners who manifest less honest behavior at the beginning of the social interaction.

On the neural level, loneliness might impact impression formation and social learning by modulating information encoding in brain regions key to social evaluations and behaviors. Previous work has highlighted the important role of the orbitofrontal cortex (OFC) and temporoparietal junction (TPJ) in social beliefs and social learning[25,26]. In particular, the TPJ has been shown to track belief updating about others[26,27], while the OFC is involved

in belief-consistent valuations of others[28,29]. The OFC might specifically be affected by biases related to impression formation and feelings of loneliness, as this region is involved in reappraisal of evaluation of social information based on an individual's internal states[30]. However, the brain regions that play a role in the formation of others' trustworthiness impressions and that are affected by subjective feelings of loneliness are still unknown. Specifically, despite initial evidence on the brain regions underlying trust propensity during single interactions in lonely individuals[11], the neural underpinnings of social information integration during multiple one-to-one interactions and their differential recruitment and contribution to varying cognitive biases in lonely individuals are still unexplored.

Here, we investigated the behavioral, computational, and neural mechanisms of how loneliness impairs the formation of accurate trustworthiness impressions in a sequential, social decision-making task. We first tested whether initial, negative information about a person's honest behavior and subjective feelings of loneliness induce a negativity bias in virtue of which lonelier individuals form more negative impressions of their social partners. In particular, we investigated whether more negative impressions in lonelier individuals are due to a stronger weighting placed on dishonesty when dishonesty is consistent with early behavioral signals of a partner's conduct. We then examined how such negativity bias impacts neural activity underlying the encoding of information about others' honesty and dishonesty, and how neural patterns during learning relate to both behavioral and computational mechanisms underlying biased impression formation in lonely individuals.

## Results

To investigate how loneliness impacts the formation of trustworthiness impressions, we constructed a task (Fig. 1 and Supplementary Fig. S1) in which participants (advisees) had to learn the character trait of their partners (advisers) through the partners' past honesty in advice giving. Information about the partner's current honesty (social information) was revealed after the decision to take the partner's advice (i.e., in the feedback phase) and optimal behavior in the task required participants to track their partner's honesty trial-by-trial. Thereby, participants were able to form impressions of their partners' trustworthiness. Importantly, some advisers started off with a dishonest behavior (i.e., initially dishonest advisers) while others showed to be honest at the beginning of the interaction (i.e., initially honest advisers), allowing for the formation of negative and positive first impressions, respectively. However, advisers' behavior changed over time, so that all advisers had the same degree of honesty during the task. This allowed us to investigate the impact of first impression on participants' behavior toward advisers with similar overall honest behavior. At the end of the experiment, participants' explicit trustworthiness judgments of the advisers and their subjective feelings of loneliness[31] were collected.

**Behavioral and computational mechanisms underlying biased impression formation.** We first examined primacy effects on learning processes underlying social information integration. We tested whether first impressions bias how subsequent information about the behavior of an adviser is processed.

First, participants were able to learn to distinguish the two advisers well ($F_{(1,68)} = 14.71$; $p = 0.0002$; $\eta_p^2 = 0.08$). Moreover, as can been in Fig. 2a, participants' advice-taking behavior significantly differed across advisers and blocks ($F_{(2,68)} = 6.03$; $p = 0.003$; $\eta_p^2 = 0.07$). At the beginning of the task (block 1, 48 trials), participants formed accurate first impressions of each of the advisers' behaviors (24 trials each) and adjusted their behavior

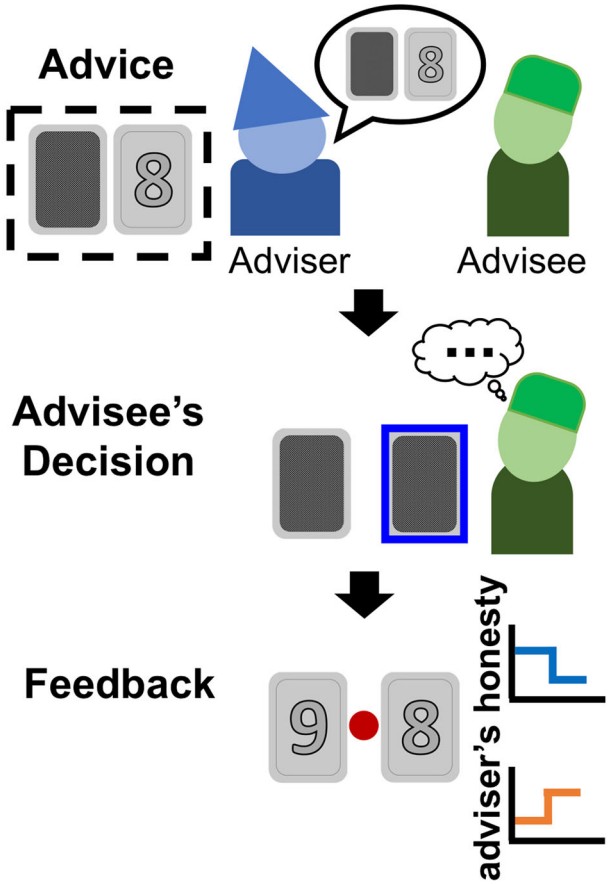

**Fig. 1 Behavioral paradigm.** Schematic representation of the take advice game. Participants in the role of advisee received advice about two covered cards from an adviser who could see one of the cards. Based on the other's advice, participants chose one of the cards and afterwards received feedback about the other's honesty (i.e., whether the adviser told the truth) and their own gains (i.e., whether they picked the higher card). The advisers' honesty changed over time, so that all advisers' behavior had the same degree of honesty.

accordingly ($t_{(34)} = 3.35$; $p = 0.002$; Cohen's $d = 0.57$). However, participants' ability to revise their behavior was strongly affected by their first impressions of the advisers in block 1. In particular, participants optimally revised their negative impressions of an adviser when that adviser began to signal honest behavior in block 2 ($t_{(34)} = 2.78$; $p = 0.009$; Cohen's $d = 0.47$). On the contrary, positive impressions loomed longer and were harder to be revised, as participants kept taking the advice of initially honest advisers even in block 2 when those advisers stopped being honest ($t_{(34)} = -1.67$; $p = 0.105$). These behavioral patterns washed out the initial impression-dependent differences in participants' advice-taking behaviors in block 2 ($t_{(34)} = 0.54$; $p = 0.594$). Finally, when at the end of the interaction (i.e., in block 3) both initially honest and dishonest advisers provided advice with the same reliability rate (so that participants should have been indifferent in choosing one over the other), we observed a strong effect of first impressions on participants' choices, such as they preferred to take advice from advisers of whom they had formed a positive first impression ($t_{(34)} = -3.41$; $p < 0.002$; Cohen's $d = 0.58$). These effects resulted in participants taking overall less advice from advisers of whom they had an initially negative impression despite all advisers having exactly the same honesty rate ($t_{(34)} = -3.85$; $p < 0.001$; Cohen's $d = 0.65$; Fig. 2b).

To test whether these behavioral patterns could be traced back to an asymmetry in the weighting of positive and negative information about the honest and dishonest behaviors of the advisers in a fashion consistent with participants' first impressions, we formally characterized our participants' behavior using computational modeling (*Methods*). In particular, we aimed at capturing how participants updated their beliefs about their partners' honesty and dishonesty and how such an updating was influenced by participants' first impressions of the advisers[32,33].

We observed that participants weighted information about the advisers' current honest and dishonest behaviors in an impression-dependent fashion ($F_{(1,34)} = 11.01$; $p = 0.001$; $\eta^2 = 0.08$; Fig. 2c). In particular, negative first impressions led to a negativity bias (i.e., the difference between the weighting on negative and positive information about an adviser's behavior), that is, stronger valuation of dishonest than honest behavior for initially dishonest advisers ($t_{(34)} = -2.74$; $p = 0.010$; Cohen's $d = 0.46$). For positive first impressions of initially honest advisers, we observed an opposite but weaker pattern indicating stronger valuation of honest behavior ($t_{(34)} = 2.07$; $p = 0.047$; Cohen's $d = 0.35$). Importantly, dishonest behavior was weighted significantly more if first impressions of the adviser were negative as opposed to positive ($t_{(34)} = 2.30$; $p = 0.028$; Cohen's $d = 0.39$). On the contrary, there was only a trend for valuation of honest behavior as a function of initially positive impressions ($t_{(34)} = -1.94$; $p = 0.061$). Hence, these results indicate a negativity bias with stronger updating of negatively-valenced behaviors in a fashion consistent with one's negative first impressions of a partner's conduct.

**Loneliness worsens existing, impression-dependent negativity biases.** We then turned to analyze participants' explicit trustworthiness judgments about the advisers. First, despite similar degrees of honesty in the game, participants' explicit reports about advisers' trustworthiness were in line with their first impressions, manifesting more negative trustworthiness judgements about initially dishonest advisers ($t_{(34)} = -2.37$; $p = 0.024$; Cohen's $d = 0.40$; Fig. 2d). This finding indicates a strong effect of first impressions on participants' beliefs about the social behavior of their partners. Consequently, participants' advice-taking behaviors correlated with their subsequent trustworthiness judgments, corroborating the evidence that participants behaviors closely aligned with their first impressions of the advisers (initially honest advisers: $r_{(33)} = 0.36$; $p = 0.031$; initially dishonest advisers: $r_{(33)} = -0.44$; $p = 0.008$; Fig. 2e).

Second, we tested whether lonely individuals had more negative expectations of their partners and whether they manifested an even stronger negativity bias induced by negative first impressions of their partner's trustworthiness. If lonelier individuals had more negative expectations of others, greater feelings of loneliness should favor the weighting of negative information especially about those of whom participants had formed more negative first impressions.

First, our data show indeed that lonelier individuals reported to have more negative expectations of others' trustworthiness ($r_{(31)} = -0.35$; $p = 0.045$; Fig. 3a). Second, loneliness was associated with a stronger negativity bias for advisers of whom participants had formed negative first impressions in the first block of the task ($r_{(31)} = 0.43$; $p = 0.012$; Fig. 3b) but not positive ones ($r_{(31)} = -0.15$; $p = 0.392$). In particular, higher levels of loneliness correlated with a stronger weighting on negative information about initially dishonest advisers than on negative information about initially honest advisers ($r_{(31)} = 0.47$; $p = 0.006$). Computationally, we observed that lonelier individuals more strongly weighted the dishonest (as opposed to the honest) behavior of initially dishonest advisers ($t_{(2,31)} = 2.20$;

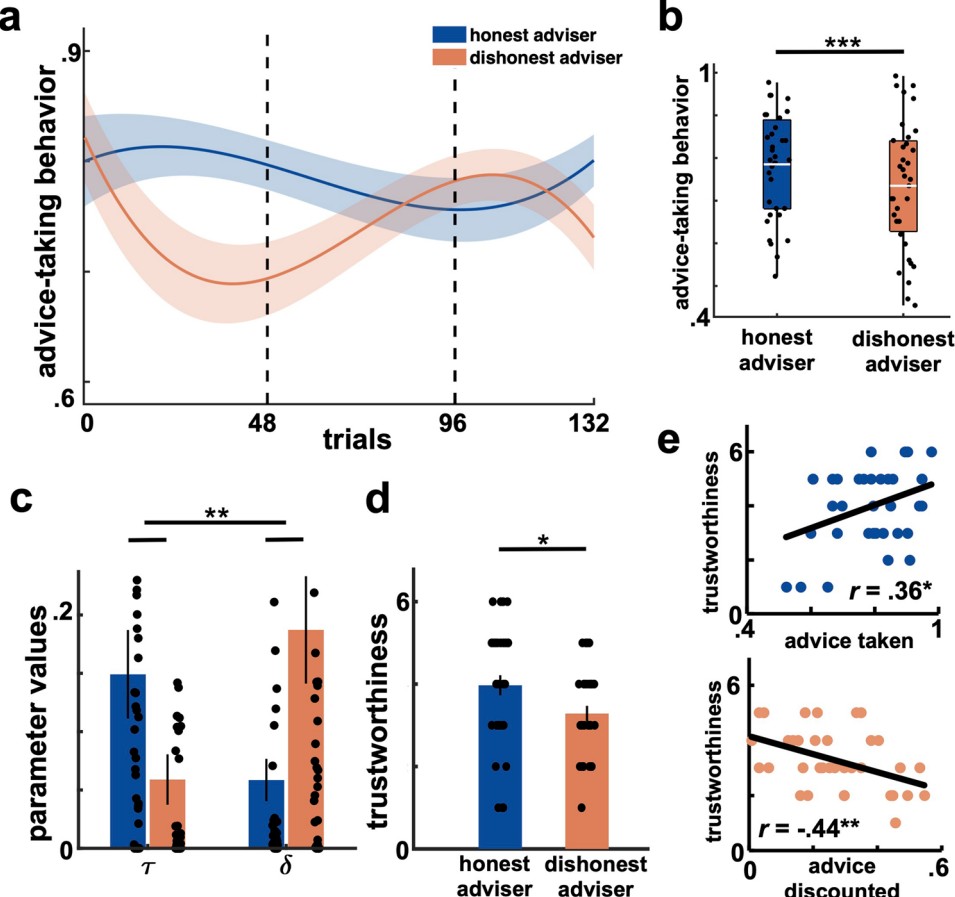

**Fig. 2 Behavioral results. a** Advice-taking behavior differed across advisers and blocks. In block 1, participants formed first impressions of the advisers based on their initially honest/dishonest behaviors. After advisers' behaviors reversed in block 2, participants' behaviors manifest a successfully revision of their negative first impressions but not of their positive ones. Finally, when advisers' behaviors had similar degrees of honesty in block 3, participants reversed to impression-dependent behavioral patterns, taking more advice from advisers of whom they had formed positive first impressions. Dashed lines separate blocks. **b** Average advice-taking behavior for each participant and adviser. **c** Impression-dependent bias in positive ($\tau$) and negative ($\delta$) information weighting for the adviser of whom participants had positive (blue, right) and negative (orange, left) first impressions. **d** Impression-dependent differences in trustworthiness judgments. **e** Correlations between participants' advice-taking behavior and trustworthiness judgments of the advisers. Top plot in blue shows the correlation for initially honest advisers, while the bottom plot in orange shows the correlation for initially dishonest advisers. In the figure, 'honest adviser' is short for 'initially honest adviser' and 'dishonest adviser' for 'initially dishonest adviser'. $\tau$, trust learning rate; $\delta$, distrust learning rate. ***$p < 0.001$; **$p < 0.01$; *$p < 0.05$. Error bars depict standard errors of the mean.

$p = 0.036$; Cohen's $d = 0.37$). On the contrary, lonelier individuals didn't more strongly weight the dishonest (as opposed to the honest) behavior of initially honest advisers ($t_{(2,31)} = -1.07$; $p = 0.292$). These results indicate that loneliness is not only associated with more negative expectations and evaluations of social partners but also biases learning patterns as to enhance the primacy effect of first impressions, especially if these were negative.

**Reduced OFC activity correlates with stronger negativity bias and greater loneliness.** Our behavioral and computational results showed how first impressions and subjective feelings of loneliness interact during impression formation about others' trustworthiness. Negative first impressions were associated with a negativity bias that was stronger in lonelier individuals. We next turn to investigate the neural underpinnings of the cognitive dynamics underlying such impression formation processes. In particular, we reasoned that failures of properly encoding positive surprise signals (a signature of better-than-expected outcomes) might hinder the encoding of more positive information about others'

behavior that could lead to the formation of more positive impressions, ultimately promoting a stronger negativity bias.

To this aim, we first examined the neural signatures tracking updates of advisers' honest and dishonest behaviors on a trial-by-trial basis by using model-based social surprise signals (see Eq. (3) in Methods). We found that activity in lateral prefrontal (e.g., dorsolateral prefrontal cortex) and parietal (e.g., inferior parietal lobule) regions tracked negative surprise signals, while activity in bilateral TPJ and OFC tracked positive surprise signals (Fig. 3c & Supplementary Table S1). Importantly, neural activity in the OFC and caudate was negatively associated with the impression-induced negativity bias, indicating that more negative first impressions of an adviser correlated with reduced OFC activity during the encoding of information about that adviser (Fig. 4a & Supplementary Table S2). This suggests that OFC activity during learning is central to the formation of accurate social impressions.

To better characterize the functional role of the OFC in this biased information processing, we more closely examined the nature of the OFC representations. Evidence that the OFC reflects the overall value of choices, options and stimuli[34] suggests that in a social interaction, the OFC might reflect an individual's first

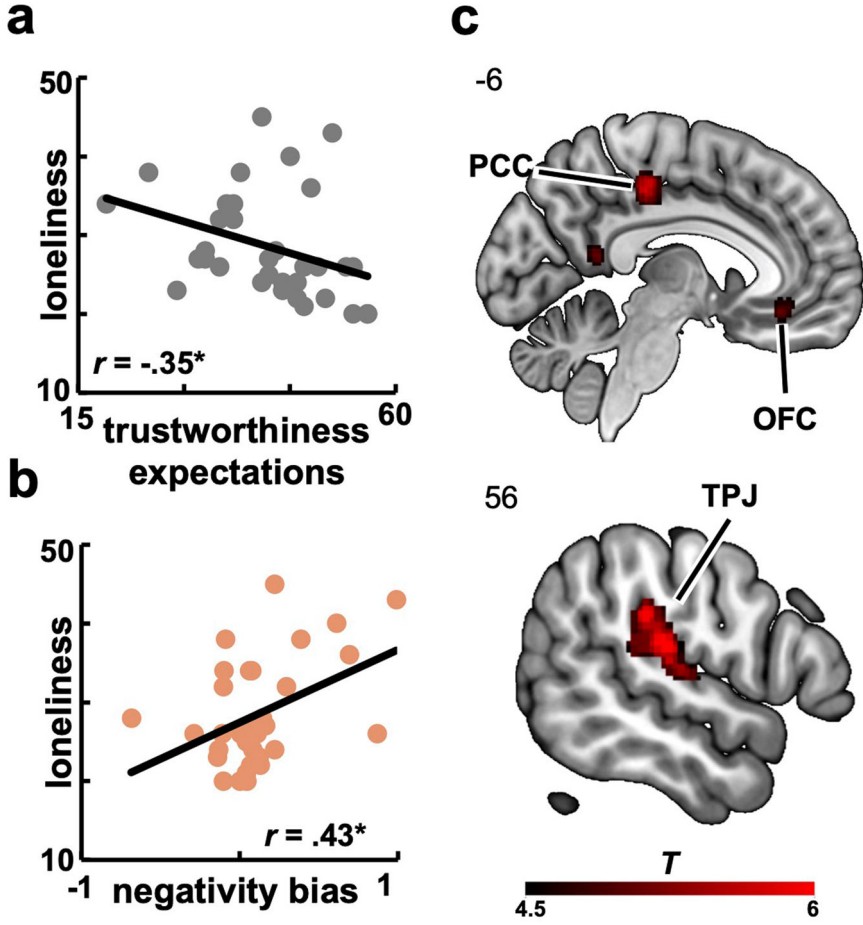

**Fig. 3 Loneliness, the negativity bias and neural surprise signals. a** Loneliness correlated negatively with general expectations of others' trustworthiness. **b** Loneliness correlated positively with the negativity bias for initially dishonest advisers. **c** Neural correlates of social surprise signals in bilateral TPJ, OFC, and PCC (*cFWE* < 0.05, cluster-forming voxel threshold *p* < 0.001). TPJ temporoparietal junction, PCC posterior cingulate cortex, OFC orbitofrontal cortex, *cFWE* whole-brain, cluster-level familywise error corrected. Heatmap represents *t* values.

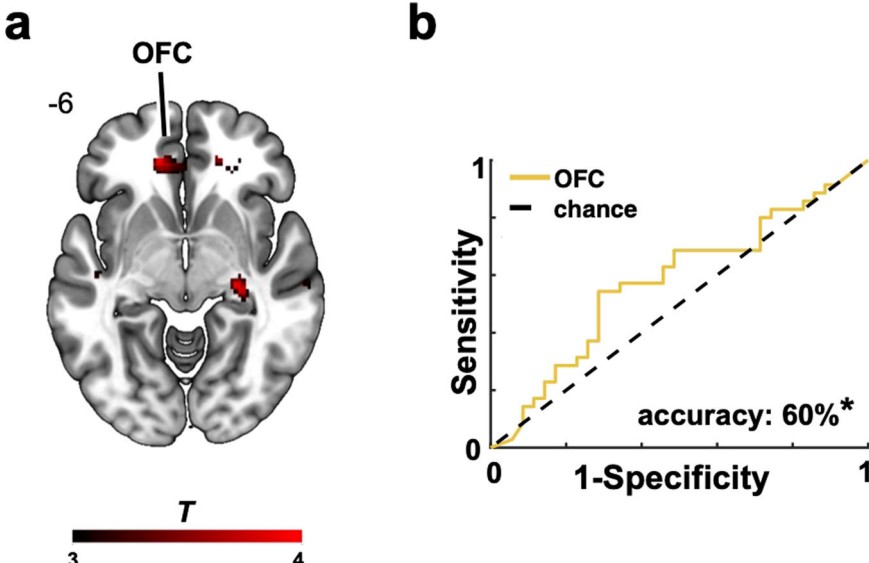

**Fig. 4 Neural correlates of the negativity bias. a** A stronger impression-induced negativity bias was associated with reduced OFC activity during learning (*cFWE* < 0.05, cluster-forming voxel threshold *p* < 0.005). **b** Neural signals in the OFC decoded participants' first impressions of the advisers. Cross-validation was based on a 20% left-out procedure and significance on 10,000 permutations. Better performance in the classification analysis is represented by higher accuracy values (more hits). OFC orbitofrontal cortex, *cFWE* whole-brain, cluster-level familywise error corrected. Heatmap represents *t* values.

impressions of their social partners. However, given its role in representations of one's and other's social traits and preferences[35,36], the OFC might as well track the learnt characteristics of a social partner. Hence, we set out to test whether the OFC in our task was more informative of participants' first impressions of the advisers or their judgments about the advisers' specific trustworthiness levels. We found that the OFC activity that correlated with a stronger negativity bias significantly classified the advisers according to their initial honest/dishonest behavior (accuracy = 60%, $p = 0.015$; Fig. 4b), but it did not predict participants' judgments about the advisers' trustworthiness (standardized mean squared error = 1.76, $p = 0.975$).

These results indicate that OFC activity reflects social information integration in a manner that is consistent with a person's first impressions. This raises the question as to whether OFC activity further reflects the stronger impression-dependent negativity bias observed in lonely individuals. As lonely individuals judged initially dishonest advisers as less trustworthy than initially honest advisers despite all advisers having the same degree of honesty, they might have engaged the OFC less optimally during the encoding of information about those advisers. To test this, we performed a whole-brain regression analysis with loneliness scores predicting neural activity encoding social information about the advisers. Our analyses confirmed that greater subjective feelings of loneliness correlated with a stronger reduction of OFC activity during social information encoding for initially dishonest advisers (Fig. 5a). Importantly, the OFC cluster from this analysis overlapped with the OFC cluster that we previously observed to correlate with a stronger negativity bias (Fig. 4a). These results suggest that the OFC plays a pivotal role in social information integration for impression formation and that modulations of neural activity in this brain area are responsible for learning biases underlying lonely individuals' more negative impressions of their social partners.

Finally, given the previously observed importance of the TPJ in tracking belief updating and of the caudate in the negativity bias, and due to the relevance of these brain regions in the literature on social learning and trust behaviors, we further tested with follow-up analyses their potential associations with individual feelings of loneliness and, consequently, the relative robustness of the observed relationship between OFC activity and loneliness. Bayesian models were run with neural activity from the OFC, TPJ and caudate during information encoding predicting subjective feelings of loneliness. These analyses confirmed the strong negative relationship between loneliness and the OFC

($\beta = -0.48$ (.16), 89% *high posterior density interval (HDI)* [$-0.75$, $-0.22$]) and revealed a weaker association between loneliness and the caudate ($\beta = -0.33$ (.18), 89% HDI [$-0.62, -0.06$]), while no significant relationship was observed between loneliness and the TPJ ($\beta = -0.02$ (.19), 89% HDI [$-0.31, 0.28$]). Importantly, when the three predictors were entered in the same multivariate model to control for the effects of the others, most of the variance explained by the caudate was explained away by the OFC, which remained the only significant predictor (OFC: $\beta = -0.49$ (.24), 89% HDI [$-0.89, -0.11$]; caudate: $\beta = -0.01$ (.25), 89% HDI [$-0.38, 0.40$]; TPJ: $\beta = -0.02$ (.17), 89% HDI [$-0.29, 0.27$]). These additional findings strengthen the importance of reduced OFC activity in belief updating for impression formation in loneliness, indicating that among the brain regions that played a role in social information encoding and the negativity bias in social learning in our study, the OFC was the one most central to subjective feelings of loneliness.

**OFC mediates the relationship between loneliness and the negativity bias.** As these findings provide convergent evidence that negative impressions are linked to reduced activity in the OFC, which was associated with both a stronger negativity bias and greater subjective feelings of loneliness, we next set out to directly test whether the relationship between loneliness and the observed negativity bias is mediated by a less engagement of the OFC. To do so, we ran a mediation analysis with bootstrap by using the OFC as mediator of the relationship between loneliness and the negativity bias. Neural activity for the mediation analysis was extracted from the OFC brain area derived from the overlap of the two OFC clusters that were independently identified in the previous two analyses on the relationships of OFC activity with the negativity bias and loneliness. We first checked whether this overlapping region in the OFC maintained strong associations with both the negativity bias and loneliness (a prerequisite for a mediation test). Correlation analyses confirmed that reduced activity in this new OFC cluster was associated with a stronger negativity bias ($r_{(31)} = -0.55$; $p = 0.001$) and greater feelings of loneliness ($r_{(31)} = -0.48$; $p < 0.005$).

Results of the mediation analysis demonstrate that when participants had a negative impression of an adviser, reduced activity in the OFC fully mediated the relationship between their subjective feelings of loneliness and the resulting negativity bias for that adviser (overall effect c: $\beta = 0.02$; SE = 0.01; $p < 0.006$; direct effect after controlling for the mediator c': $\beta = 0.01$; SE = 0.01; $p = 0.491$; Fig. 5b). Confirming our previous correlation tests, the OFC entertained significant negative

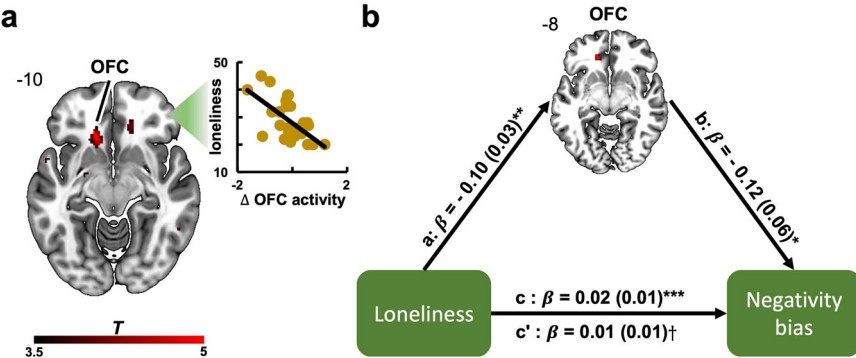

**Fig. 5 OFC mediates the effects of loneliness on the negativity bias. a** Greater feelings of loneliness were associated with reduced OFC activity when receiving feedback from an advisor participants had negative first impressions of ($x,y,z = -14,26,-10$; $cFWE_{svc} < 0.05$). **b** Decrease of OFC activity fully mediated the effects of loneliness on the negativity bias. The mediation analysis was performed using bootstrap test with 10,000 permutations. Standard errors in parentheses. OFC orbitofrontal cortex, $cFWE_{svc}$ small-volume, cluster-level familywise error corrected. Heatmap represents $t$ values. ***$p < 0.005$; **$p < 0.01$; *$p < 0.05$; † nonsignificant.

relationships with loneliness (effect a: $\beta = -0.10$; SE = 0.03; $p < 0.01$) and the negativity bias (effect b: $\beta = -0.12$; SE = 0.06; $p < 0.05$) in the mediation analysis. These results suggest that the less the OFC is engaged during integration of information about others' behaviors, the stronger the relationship between feelings of loneliness and the negativity bias in learning, providing evidence on the neurocomputational underpinnings of biased social learning in lonely individuals.

**OFC-TPJ coupling underlies more positive trustworthiness beliefs.** Our results so far have suggested that a less efficient information processing in the OFC is associated with the formation of more negatively-biased impressions. However, because there were no differences among the advisers in their actual honest behaviors, a better information integration could have compensated for this negativity bias, leading to more favorable impressions of the advisers who were initially dishonest. Following this line of reasoning, we investigated functional pathways between the OFC and other brain regions that could have supported a better integration of information about the advisers' behaviors.

A whole-brain functional connectivity analysis revealed that the OFC was functionally coupled to the left TPJ during encoding of social information about the advisers' behavior (Fig. 6a). Importantly, stronger coupling between the OFC and left TPJ was associated with more positive trustworthiness judgments for the advisers who were initially dishonest ($r_{(33)} = 0.34$; $p = 0.045$; Fig. 6b) but not for those who were initially honest ($r_{(33)} = 0.05$; $p = 0.761$), suggesting a potentially compensatory mechanism

underlying the integration of behavioral information that could promote the formation of less biased social beliefs. Importantly, this relationship between stronger OFC-TPJ coupling and more positive trustworthiness judgments about initially dishonest advisers was moderated by participants' feelings of loneliness. Specifically, a moderation analysis revealed that this relationship was stronger for higher levels of loneliness ($\beta = 0.30 (0.15)$, 89% $HDI$ [0.04, 0.57]; Fig. 6c), suggesting that a stronger coupling between the TPJ and OFC was associated with more positive trustworthiness judgments particularly in lonelier individuals. This further lends support to the hypothesis that such coupling might represent a buffering mechanism against the negative effects of loneliness on impression formation and learning.

**Discussion**
In this study, we showed that first impressions have a long-lasting effect on people's beliefs about others' trustworthiness and that these impressions impact how people integrate social information about others via modulation of OFC activity. In particular, lonely individuals did not only have more negative expectations of others, but they also manifested a stronger negativity bias that was accompanied by reduced neural activity in the OFC. Such reduced OFC activity mediated the relationship between loneliness and the negativity bias, leading to less positive trustworthiness beliefs about others' social behaviors. However, stronger functional coupling between the OFC and TPJ (which tracked belief updating from positive behavioral information) was associated with more favorable trustworthiness judgments, especially in lonelier individuals, indicating a possible compensatory

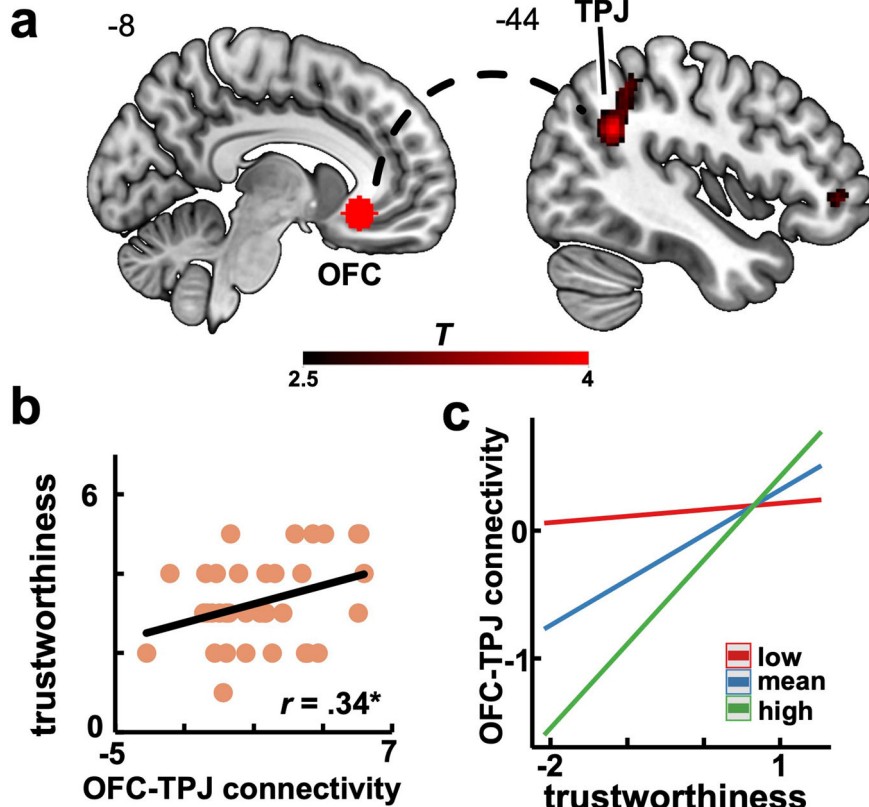

**Fig. 6 OFC-TPJ functional coupling. a** A connectivity analysis revealed a functional coupling between OFC and TPJ during information encoding ($cFWE_{svc} < 0.05$). **b** Stronger OFC-TPJ functional connectivity correlated with more positive trustworthiness judgments about the initially dishonest adviser. **c** Moderation effect of loneliness levels (regression lines) on the relationship between the OFC-TPJ functional connectivity and trustworthiness judgments of the initially dishonest adviser (using standardized regressors). OFC orbitofrontal cortex, TPJ temporoparietal junction, $cFWE_{svc}$ small-volume, cluster-level familywise error corrected. Heatmap represents $t$ values.

mechanism for the negativity bias in information processing and impression formation.

First, behaviorally we observed that positive first impressions loom longer than negative ones and were harder to revise, leading to biased trustworthiness judgments about initially dishonest advisers even though their overall conduct was as honest as that of initially honest advisers. This is contrary to previous studies suggesting that negative first impressions loom longer than positive ones and are less likely to be revised[17,37] but consistent with more recent work[21,22]. One possibility for these different results in the literature might be traced back to the different types of experimental paradigms employed. Recent work showing that negative first impressions are more variable and easier to revise uses interactive social paradigms in which participants must learn trial by trial the character traits of their interacting partners and form an overall impression of them via experience. On the contrary, older studies have mostly used descriptions or elicitation methods with no or little learning by experience[15,38–40]. Hence, similar to studies in the risk literature[41,42], also social tasks might have suffered from the "description-experience gap" with different effects of first impressions on social behaviors depending on whether impressions were formed via experience or via description.

Further, these behavioral results suggest that first impressions might lead to an asymmetry in information weighting and updating consistently with an individual's first impressions. Our computational modeling results demonstrated that participants were integrating behavioral signals from the advisers in a manner consistent with their initial impressions of those advisers. Particularly, dishonest behavior was weighted significantly less for initially honest advisers, suggesting that the biased trustworthiness judgments observed on the behavioral level might be due to a biased integration of social information about the advisers on a trial-by-trial basis. These results suggest that positive first impressions loom longer because they impact participants' ability to accurately integrate impression-inconsistent information for belief update and provide a computational account for the mechanisms underlying (un)biased impression formation in interactive social contexts.

Importantly, we found that the greater weight on dishonest behavior for initially dishonest advisers scaled as a function of subjective feelings of loneliness. Loneliness has been hypothesized to act as a warning signal that informs individuals about their unsatisfactory social bonds and prompts them to seek meaningful relationships[5,43]. With this respect, loneliness is a healthy evolutionary tool to weigh the supportiveness of one's social network and improve it if needed. However, other evidence has suggested that loneliness is associated with hypervigilance for negative social cues that likely leads to negatively biased evaluations of social interactions[8,13], explaining social withdrawal and inflated fear of social judgments in lonely adults[44–46].

In our study, we show not only that lonelier individuals have more negative general expectations of others' trustworthy behavior but also that they manifest a stronger negativity bias with more negative trustworthiness beliefs about their social partners. Modeling results showed that lonelier individuals more strongly integrate negative information about their partners if their first impression of the partner is negative. Hence, these results indicate that loneliness is characterized by biased learning dynamics leading to the formation of more negative impressions and evaluations of others, which on the long run might give rise to lonely individuals' more negative expectations of their social partners. Importantly, these learning patterns in lonelier individuals are intriguingly similar to those previously observed in depression[47], and given that loneliness has been observed to be one of the best predictors of the onset of depressive symptoms[48], our results

suggest that the observed, loneliness-induced asymmetry in processing positive and negative feedback could represent an early biomarker for later development of depression.

On the neural level, this asymmetry in information processing was reflected by reduced OFC activity during social information encoding (i.e., when participants received feedback about the past honest behavior of their partners). As this brain region further tracked positive surprise signals, these results suggest that a reduced recruitment of the OFC for encoding of positive information about social partners might play a role in the formation of more negative social beliefs. Importantly, activity in the OFC was observed to be more informative of an individual's overall impression of a partner rather than the encoding of the learnt trustworthiness of the advisers, suggesting that OFC activity preferentially represents general trait information about others (possibly along a valence dimension) consistent with one's first impressions.

This reduced OFC activity during social information encoding was further negatively associated with subjective feelings of loneliness with lonelier individuals showing less OFC engagement. Further, such reduced engagement of the OFC mediated the negative relationship between loneliness and the negativity bias, indicating that the OFC represents trait characteristics of another person in line with an individual's first impressions and internal psychological loneliness states. This aligns with the general hypothesis that the OFC guides behavioral responses by representing the specific identity of a stimulus based on the agent's current internal psychological state[49]. Importantly, reduced functional engagement of the OFC and anatomical abnormalities in this brain area have previously been associated with depression as well[50,51], corroborating the evidence that loneliness might be employed as both a behavioral/computational and neural marker of depression onset.

Importantly, a stronger coupling was observed between the TPJ and the OFC during the encoding of information about initially dishonest advisers and this coupling was related to more positive trustworthiness judgments of such advisers. Given its importance as associative brain area[52] and its relation with positive surprise signals in our study, the TPJ might have a role in tracking changes in honest and dishonest behaviors of the advisers for belief updating and impression revision[53,54]. In particular, the coupling with the OFC and its relationship with more favorable trustworthiness judgments suggest that this brain region likely contributed to a more accurate integration of positive information about a social partner, leading to the formation of more positive impressions.

Moreover, we observed that this relationship was moderated by individual feelings of loneliness. Specifically, for higher loneliness levels, the coupling between these two brain regions more strongly predicted positive trustworthiness judgments of initially dishonest advisers. These converging results suggest that the impairment of a more accurate revision of an individual's initial negative impressions due to decreased OFC activity during the encoding of new incoming information could have been buffered by a more efficient information exchange between the OFC and the TPJ[55]. Hence, this functional pathway might reflect a compensatory mechanism for negativity biases in information weighting and impression formation, particularly in lonely individuals.

A potential limitation of this study is that we did not manipulate subjective feelings of loneliness. However, we believe loneliness should be treated like other clinical disorders such as depression and anxiety[56] and hence investigated by using a combination of questionnaire-based assessments, ecological social paradigms and computational modeling to gain insights into the mechanisms put in place by lonely states, like we did in this work.

Moreover, in future work especially with bigger samples, additional, advanced statistical methods like stratified sampling with a longitudinal approach[57] should be employed to more closely isolate the operating dynamics of loneliness from other factors (e.g., personality, psychoses), test how the biases in learning and impression formation highlighted in the current work lead to the emergence of overly negative social expectations in lonely individuals, and study their temporal dynamics[58,59].

Taken together, this work provides valuable insights into the multifaceted dynamics underpinning impression formation during social interactions. We demonstrated how the interaction between first impressions and subjective feelings of loneliness lead to more negative beliefs about others via modulation of neural activity in the OFC, thereby shedding light on the psychological, computational and neural mechanisms underlying the emergence of biased social evaluations and expectations. Importantly, this study provides evidence for loneliness as a promising behavioral and neural marker for predictions of the onset of clinical symptoms such as depression with the potential of new research avenues for basic and clinical investigations.

## Methods

### Experimental procedures

*Subjects*. Thirty-five participants (25 females) participated in the experiment (age: $22.37 \pm 2.62$ $M \pm SD$). Participants were recruited from the student community at the University. They were all right-handed and had no history of neurological or psychiatric disorders. Participants gave written informed consent after a complete description of the study was provided. The study was approved by the ethics committee of the University of Luebeck and all ethical regulations relevant to human research participants were followed.

*Take advice game*. In the take advice game (TAG), participants played as advisee a card game with two other co-players who received the role of adviser. This game is a sequential decision-making game played with each of the advisers over multiple trials, allowing participants to receive feedback about each adviser's advice-giving behavior in every trial and learn about the advisers' honesty trial-by-trial. This one-to-one interaction with the advisers enabled participants to form impressions of the advisers and adapt their own subsequent trusting behavior accordingly. Participants and their co-players were invited to the lab and instructions about the experiment were provided. They were instructed that the roles in the game were randomly assigned. For role assignment, participants and their co-players needed to draw a ball from a lottery box before starting the experiment. Further, to guarantee anonymity during the experiment, participants and their co-players needed to choose an avatar that represented themselves at the beginning of the game (Supplementary Fig. S1). In truth, participants always received the role of advisee.

As advisee, participants' task was to pick one of two cards and try to draw the card with the highest number to win money. Card numbers ranged from 1 to 9 (except for 5). For their decisions, participants relied on the advice provided by the advisers (establishing an adviser-advisee interdependency necessary for trust). The advisers saw one of the two cards (adviser phase: 2–3 s) and communicated the card number to the participants (advice phase: 1 s). Thus, participants were aware that the advisers knew more than them but did not have complete knowledge (i.e., they did not know which card was the winning one). Hence, the advice was not about the best decision to make but represented only additional information to help participants make their decision. Such experimental setting is similar to real-life scenarios in which people seek more knowledgeable

individuals for advice, who however rarely have complete knowledge about any particular life circumstance and hence rarely can tell what the optimal decision to make is. Finally, participants chose one of the two cards (decision phase: 1 s) and received feedback (feedback phase: 1 s). In the feedback phase, participants were provided with social information (card numbers were informative of the adviser's honesty in advice giving) and nonsocial information (green and red circles represented winnings and losses, respectively).

Advisers' honest behavior was probabilistic and changed over the experiment. There were 3 blocks in total. In the first, impression-formation block, one adviser was honest 75% of the time and dishonest the rest of the time, whereas the other adviser was honest 25% of the time and dishonest the rest of the time. Here, participants could form first impressions of the advisers' honesty and dishonesty in advice giving via trial-by-trial learning. Based on pilot data, we observed that, given the low informativeness of the behavioral signal from the advisers, its uncertainty, stochasticity and non-stationarity, and the fact that participants received only indirect information about the honesty of the advisers, which needed to be inferred from the correctness of the advice and not from the accuracy of their decisions, twenty-four trials were required for participants to form stable impressions of each of the advisers. Importantly, we did not provide any information about the potential rationales behind the advisers' behaviors, as we were interested in investigating how learning biases, especially the negativity bias in lonely individuals, emerge from participants' trial-by-trial learning patterns, and we thus wanted to avoid priming and experimentally inducing any bias in participants' prior expectations that could have impacted their learning and belief updating. In the second block, advisers' honest behavior reversed and in the third block, both advisers were equally honest. Here, we tested the impact of first impressions on participants' ability to update their beliefs about the advisers and their behavioral strategies. Intertrial stimulus intervals (ISIs) were 2–8 (mean: 2.6 s) seconds long. Jitters between trials were 2–8 (mean: 4 s) seconds long. Participants played a total of 4 runs (i.e., fMRI scanning sequence) in the scanner with 66 trials each for a total of 264 trials. Participants used a standard MRI-compatible button box to make their choices in the MRI scanner. Participants had small breaks between one run and the other during which they laid in the MRI scanner and were instructed to keep still.

*Trustworthiness ratings*. After the scanning session, participants rated each adviser's trustworthiness in the TAG. Ratings followed on a 7-point Likert-scale from very untrustworthy to very trustworthy. These ratings measured participants' explicit trustworthiness judgments about the advisers from their interactions in the TAG.

*Questionnaires*. To acquire data on subjective feelings of loneliness and the participants' general trustworthiness expectations, participants filled out two questionnaires at the end of the experiment. For subjective feelings of loneliness, participants completed the UCLA loneliness scale[31,60]. Loneliness scores ranged from 20 to 45 with a mean of 28.52 (median = 26; SD = 6.84). For general trustworthiness expectations, participants completed the preference survey module for trust preferences[61]. Scores ranged from 19 to 56 with a mean of 42.06 (median = 43; SD = 8.98). Due to technical problems, the first two participants did not complete these questionnaires, leaving a total of 33 participants for analyses of questionnaire data. Finally, in an open question at the end of the experiment, participants were asked to report whether they believed they were playing with other participants during the game. To avoid social desirability effects, we told participants that consistently with the cover story at the

beginning of the experiment, we had to use avatar during the game due to anonymity reasons and that now, for statistical purposes, we were interested in knowing whether they genuinely thought they were playing with the other participants during the game. Two independent raters coded participants' written responses. Results show that about 78% of our participants believed they were playing with others.

### Scanning parameters and preprocessing

*Image acquisition.* Data were collected with a 3-Tesla Siemens MAGNETOM Skyra whole-body MRT-scanner equipped with a 64-channel sensitivity-encoding head coil. The fMRI scans consisted of approximately 900 contiguous volumes per run (axial slices, 56; slice thickness, 3 mm; no interslice gap; TR, 1000 ms; TE, 30 ms; acceleration factor, 4; flip angle, 60°; voxel size, $3.0 \times 3.0 \times 3.0$ mm$^3$; FOV, $204 \times 204$ mm$^2$). High-resolution structural images were acquired through a 3D sagittal T1-weighted MP-RAGE (sagittal slices, 208; TR, 2300 ms; TE, 2.43 ms; slice thickness, 0.85 mm; voxel size, $0.85 \times 0.85 \times 0.85$ mm$^3$; flip angle, 8°; inversion time, 1100 ms; FOV, $240 \times 240$ mm$^2$).

*Image preprocessing.* Neuroimaging data analyses were performed on SPM12 (v. 6905; http://www.fil.ion.ucl.ac.uk/spm/software/spm12/) in MATLAB 2019a (The Mathworks, Natick, Massachusetts; http://www.mathworks.com/). The functional images were corrected for slice acquisition time and voxel displacement using field maps, realigned for head movement correction to the mean image, co-registered to their structural images using the unified segmentation procedure[62], normalized into MNI space using deformation fields from the segmentation procedure (resampling voxel size: $2 \times 2 \times 2$ mm$^3$), and spatially smoothed using a Gaussian filter ($8 \times 8 \times 8$ mm$^3$ full width at half maximum, FWHM) to decrease spatial noise.

### Analyses

*Behavioral analyses.* A 3 (block) × 2 (adviser) repeated-measures ANOVA was computed to test differences in advice-taking behaviors toward advisers across blocks. T-tests were used to test for statistically significant differences in trustworthiness ratings between advisers. To test the associations between the OFC, TPJ, and caudate activity and subjective feelings of loneliness, we fitted Bayesian regression models in Stan (https://mc-stan.org) with standardized regressors using the brms[63] package in R, which employs the no-U-turn sampler for efficient exploration of posterior estimates. We ran 8 chains with 21,000 iterations each and 1000 burn-in samples using uninformative priors ($\mathcal{N}(\mu, \sigma^2)$ where $\mu = 0$ and $\sigma^2 = 10$). Posterior point estimates represented the expected value of the posterior distributions of models' parameters. Uncertainty in the estimation of the models' parameters was represented by 89% highest posterior density intervals (HDI).

*Computational models.* We tested different computational models and used random-effects Bayesian model comparison to select the winning model[64–66]. For each model, the Akaike Information Criterion (AIC) was computed as measure of model evidence, which represents a trade-off between accuracy (the model log likelihood) and complexity (the number of free parameters to estimate):

$$AIC = -2 * LL + 2 * np, \tag{1}$$

where *np* refers to the total number of free parameters to estimate and *LL* to the log likelihood of the model, which was computed

for each participant as follows:

$$LL = \sum_i \log(P(Data_i|Model)) \tag{2}$$

Based on the log model evidences (i.e., -AIC/2), the exceedance probability for each model was estimated (using *spm_BMS*), that is, how likely it is that any given model is the most frequent in the model space considered[67]. Random-effects model selection based on the exceedance probability is superior to fixed-effects model selection procedures that average model evidences over individuals (such as log group Bayes factor) because it does not assume that individuals are sampled from a homogenous population with one (unknown) model and is not biased by individual outliers (e.g., a model that is extremely good only for one or a couple of individuals in the sample).

We tested three classes of models. One class of models did not distinguish between the type of social information (honest or dishonest) and updated the subjective value of trusting the adviser either independently from nonsocial information (winnings or losses) or separately for each type of nonsocial information using one single learning rate (*M1-3*) or two different learning rates (*M4*). Another class of models, on the contrary, distinguished between the type of social information and updated subjective values irrespective of the received nonsocial information with one (*M5-6*) or two learning rates (*M7*). A last class of models consisted of the combination of the first two and updated subjective values by distinguishing the type of both social and nonsocial information with two (*M8-9*) or four learning rates (*M10*).

Random-effects Bayesian model comparison (Supplementary Fig. S2) indicated that the winning model was a model with two learning rates (*M7*) weighting the type of social information separately, as follows:

$$V_t = V_{t-1} + \tau S_t \mathbf{1}_t + \delta S_t (1 - \mathbf{1}_t) \tag{3}$$

where $V_t$ is the subjective value of trusting the adviser on trial $t$, $\tau$ is the honesty learning parameter, $\delta$ is the dishonesty learning parameter and $S_t$ is the social surprise signal (i.e., $I_t - V_{t-1}$, where $I_t$ is the type of social information received on trial $t$). Importantly, both $\tau$ and $\delta$ were estimated separately for each of the advisers. An individual's negativity bias was operationalized as the difference between $\delta$ and $\tau$, pointing to the imbalanced weighting between positive and negative information about the adviser. Finally, $\mathbf{1}_t$ is an indicator function for the honesty of the received social information taking the following values:

$$\mathbf{1}_t = \begin{cases} 1 & \text{if social information was honest} \\ 0 & \text{if social information was dishonest} \end{cases}$$

Trial-by-trial subjective values were transformed into trust probabilities using the following stochastic decision rule (i.e., softmax function):

$$p_t = \left(1 + e^{-\beta V_t}\right)^{-1}, \tag{4}$$

where $p_t$ is the probability of choosing to trust at time $t$ and $\beta$ is the participant-specific inverse temperature–a free parameter capturing noise in participants' choice behavior.

*First-level neuroimaging analyses.* On the first level, a general linear model (GLM) was estimated for each run with parametric modulators of the feedback phase containing model-based, trial-by-trial surprise estimates. For each task phase, two regressors were estimated (one for each adviser). For the feedback phase, a parametric modulator entailing trialwise surprise values was added. Moreover, motion parameters were included as regressors of no-interest. A temporal high-pass filter with a cutoff of 128 s was applied. Contrast analyses between beta regressors were

performed on the first level and later used for second-level whole-brain analyses.

*Second-level neuroimaging analyses.* To examine the neural signatures of social surprise (model-based S estimates from Eq. (3)), a one-sample t-test on the parametric modulators of the feedback phase was performed at the second (group) level. To investigate the relationships between neural responses to social information and the impression-induced negativity bias, whole-brain contrast images of the feedback regressor were correlated with learning rate differences in a subject-level whole-brain regression analysis. Results were whole-brain corrected for multiple comparisons using a voxel-level threshold of $p < 0.001$ and a family-wise error, cluster-level ($FWE_c$) corrected threshold of $p < 0.05$[68]. Small-volume correction for the OFC was based on an independent anatomical OFC volume provided by the SPM Anatomy toolbox (v. 2.2)[69].

To test the relationships between neural responses to social information and subjective feelings of loneliness, the same whole-brain contrast images of the feedback regressor were correlated with loneliness scores from the UCLA loneliness scale in a subject-level whole-brain regression analysis (using $FWE_c < 0.05$ with a voxel-level threshold of $p < 0.005$).

*Multivariate classification and regression analyses.* To investigate the relationships of the neural signal in the OFC with participants' impressions of the advisers and their trustworthiness judgments, multivariate classification and regression analyses were performed. Between-adviser differences of OFC neural responses to social information were used as feature. Voxelwise betas were extracted from within the cluster yielded by the previous whole-brain regression analysis with the impression-induced negativity bias. Overall impression of an adviser as honest or dishonest was used as binary target for the classification analysis. The difference in trustworthiness ratings between advisers was used as continuous target for the regression analysis.

For the multivariate classification analysis, logistic boosting regression was employed building an ensemble of 500 classification trees with a learning rate parameter = 0.01 as implemented in *fitcensemble* in MATLAB with *LogitBoot* as method. For the multivariate regression analysis, least-square boosting regression was employed building an ensemble of 500 classification trees with a learning rate parameter = 0.01 as implemented in *fitrensemble* in MATLAB with *LSBoost* as method. For both multivariate analyses, a 20% left-out cross-validation approach[70] was used where the algorithm was trained on 80% of the data and tested on the left-out 20% in each of the five folds. Cross-validated performance was tested against a permutation test with 10,000 permutations (n_perm). In each permutation, the multivariate algorithm was trained on randomly permuted labels using the same 20% left-out cross-validation procedure of the true model. The sum of models trained on permuted labels that performed better than the true model was then computed (p_models). The nonparametric $p$ value was assessed including the observed statistics according to the following formula[71]:

$$\frac{1 + p models}{1 + n perm} \qquad (5)$$

Standardized mean squared error (smse) represented the performance metrics of the multivariate prediction analysis. Better performance is represented by lower smse values. Percentage accuracy represented the performance metrics of the multivariate classification analysis. Here, better performance is represented by higher accuracy values.

*Mediation analysis.* To examine whether OFC activity mediated the effects of loneliness on the impression-induced negativity bias, average OFC beta values were computed for each participant and used as mediator in a mediation analysis with bootstrap test for statistical significance (10,000 permutations). Loneliness scores from the UCLA loneliness scale were used as independent variable, the negativity bias as dependent variable and subject-level average OFC activity as mediator. The mediation analysis was performed using the Multilevel Mediation and Moderation (M3) toolbox for MATLAB[72].

*Task-dependent functional connectivity analysis.* To investigate the potential functional pathways of the OFC, a task-dependent functional connectivity analysis was implemented using a whole-brain psychophysiological interaction analysis (PPI)[73,74] with the OFC as seed region (10 mm radius). The PPI-GLM consisted of a task regressor, a physiological regressor entailing deconvolved blood-oxygen-level-dependent (BOLD) signal from the seed region and a regressor for the interaction term. Movement parameters were entered as regressors of no interest. Significant connectivity was assessed with a voxel-level threshold of $p < 0.001$ and an FWE cluster-level threshold of $p < 0.05$ within the ROI[75].

*Moderation analysis.* To study the moderation role of individual levels of loneliness in the relationship between the OFC-TPJ functional connectivity and the trustworthiness judgments about the advisers, we run a moderation analysis with loneliness scores as moderator of trustworthiness ratings of the advisers predicting the functional coupling between the OFC and TPJ. Generally, a moderator affects the zero-order correlation between two other variables substantially reducing or reversing the relationship between these variables (Baron & Kenny, 1986). Formally, a moderator is represented by a significant interaction between a focal predictor and the moderating variable in predicting an outcome variable as follows:

$$y_i = \beta_0 + \beta_1 x_i + \beta_2 m_i + \beta_3 x_i m_i + \varepsilon_i, \qquad (6)$$

where responses $y_i$ (here, the subject-level strength of the coupling between the TPJ and OFC) are linearly predicted by the regressor $x_i$ (here, participants' judgments of the trustworthiness of the initially dishonest adviser) and this predictive relationship is moderated by the moderator $m_i$ (here, subjective feelings of loneliness). The moderation was tested with a Bayesian model in Stan using the same approach as the above-mentioned Bayesian regression models.

*Statistics and reproducibility.* Repeated-measures ANOVA was computed to test differences in advice-taking behaviors and parameter estimates between advisers. T-tests were employed for post-hoc tests and comparison of trustworthiness ratings between advisers (two-tailed). Correlation analyses were performed using linear Pearson correlations. Sample size was based on a previous behavioral study with a sample of 33 participants in which the interaction effect between advisers and blocks on advice-taking behavior had a $\eta_p^2 = 0.12$, achieving a power > 0.99, while the interaction effect between advisers' first impressions and information type (positive/negative) on model parameters had a $\eta_p^2 = 0.21$ with a power = 1[21]. Similarly, in our study, we reached an effect size $f = 0.30$ with an achieved power > 0.99 for both interaction effects on behavior and model-based information weighting. Bayesian regression analyses were run to test associations between brain signals and loneliness levels. Whole-brain general linear models and multivariate analyses were performed to test for significant brain correlates.

*Labeling and data visualization.* The SPM Anatomy toolbox (v. 2.2)[69] and MRIcron (http://people.cas.sc.edu/rorden/mricron/install.html/) were used for anatomical labeling. MRIcroGL (https://www.mccauslandcenter.sc.edu/mricrogl/home/) was used for brain visualizations.

**Reporting summary**. Further information on research design is available in the Nature Portfolio Reporting Summary linked to this article.

## Data availability

Numerical source data for the plots and graphs in the manuscript are available on OSF at https://osf.io/raf5h/. Unthresholded statistical maps were uploaded to NeuroVault.org database and are available at https://neurovault.org/collections/15181/.

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

## Acknowledgements

S.Q.P. is supported by the German Research Foundation Grants INST 392/125-1, PA 2682/1-1, PA 2682/2-1 and by a grant from the German Ministry of Education and Research (BMBF) and the State of Brandenburg (DZD grant FKZ 82DZD00302).

## Author contributions

G.B. conceived the idea. G.B. and S.Q.P. designed the experiment. G.B. collected and analyzed the data. G.B. wrote the manuscript with edits from S.Q.P. All authors approved the final version of the manuscript.

## Competing interests

The authors declare no competing interests.
