## [Peer Review File · Communications Biology]

Reviewers' comments:

Reviewer #1 (Remarks to the Author):

The current study provides very interesting insights into impression formation in lonely individuals regarding trust and honesty. The results indicate that lonely individuals show stronger negative expectations about others and a biased information updating for negative first impressions. Moreover, this negativity bias in information updating was accompanied by reduced orbitofrontal cortex activity. The study addresses a very important and insufficiently studied topic and I highly appreciate the authors approach of combining behavioral, computational, and fMRI methods. The manuscript is well-written and the study is well-designed. I only have some minor remarks which might help to improve the manuscript and data presentation:

- The authors introduce cognitive biases related to loneliness in a comprehensible manner. Nevertheless, various studies investigated the relationship of loneliness with interpersonal trust in particular (e.g., Rotenberg et al., 2010, doi: <https://doi.org/10.1177/0146167210374957>; Lieberz et al., 2021, doi: <https://doi.org/10.1002/adv.202102076>) which should be mentioned when introducing this topic. Moreover, in lines 75-77 the authors state that the neural basis of biased trust in loneliness is still unknown. However, this question was addressed by Lieberz et al. (2021) as well.
- For all reported statistical effects, effect sizes should be provided. This is especially true given the interpretation of the reported effects as "strong" (see, for example, line 124).
- Some effects are only reported in figure legends but not in the main text. This concerns OFC and TPJ activity (figure 3c/4a) but also the correlation of loneliness with OFC activity (figure 5a). I strongly suggest to report all statistical effects comprehensively in the main text as well.
- Figure 4c is missing
- Lines 194-198: The interpretation of the relationship of brain activity with the negativity bias is interpreted in a too directional manner. This is especially true as the interpreted direction of the relationship is reversed later on.
- The analysis focuses on the OFC activity which is plausible given the observed association of OFC activity with the negativity bias. However, the caudate showed a significant association with the negativity bias as well and the bilateral TPJ encoded positive surprise signals. Moreover, the TPJ was introduced as being important to track belief updating. Why wasn't the TPJ activity analyzed in a more comprehensive way with regard to loneliness? I would recommend to additionally report associations of loneliness with TPJ and caudate activity. If a relationship of loneliness with brain activity is specific for OFC activity, this might further strengthen the importance of reduced OFC activity in belief updating in loneliness.
- Take advice game: Did participants believe that the adviser was played by another participant? It is not clear to me why the adviser should lie, which could prevent the subjects from believing that it was another real person. Was there a cover story that the adviser could also increase their own outcome which might have made it necessary to lie? This is important given that the social context is mandatory to measure trust in contrast to learning in a non-social context.
- Please provide a more comprehensive description of the sample with regard to loneliness and general trustworthiness expectations.

Reviewer #2 (Remarks to the Author):

The present research tackles several important questions regarding impression formation and updating. First, the authors investigate the presence of a negativity bias in forming impressions (weighting negative first information more than positive). Second, the authors investigate a negativity bias that is especially likely to be observed among lonelier individuals. Finally, the authors investigate

a potential link between orbitofrontal cortex (OFC) activity and negativity bias, with the goal of seeing whether the OFC activity can explain the hypothesized link between loneliness and negativity bias and first impression.

To start with the strengths of the present work, I would like to applaud the researchers for raising such interesting questions and aiming to address an important gap in the literature on impression formation. Their interdisciplinary approach is valuable and innovative.

Unfortunately, however, I refrain from recommending the current work for publication. Below you can find a list of major and minor issues I observed during my review. Please note that my review excludes the details regarding the computational models as I am not an expert in that area.

Major Issues

The issue of causality: Throughout the paper, the authors claim a causal relationship between loneliness and negativity bias. However, as the present research does not include experimental manipulation of loneliness, it is not possible to infer any directionality. Technically, it is possible that individuals who happened to weigh negative information more heavily in the advice-taking task reported feeling lonelier at the time of the experiment (as a psychological state), or there could be a bidirectional relationship between these variables. As one of the biggest premises and the theoretical foundation of the present paper is the social cognitive outcomes of loneliness, an experimental manipulation (or at least a longitudinal design) would be essential for this particular research question.

Conceptual unclarity: There are a few concepts that remain unclear and potentially misleading in the writing. First, it is unclear how the authors operationalize "negativity bias" throughout the text -when are they referring to computational, and when are they referring to behavioral outcomes? I found myself completely lost trying to disentangle their results. Second, the authors cite a few studies on initial expectations in their introduction (p. 4). Then they refer to the first block in their take advice game as "expectations" or "first impressions" interchangeably (p. 10). As many rounds of interactions take place in the first round (66 -as mentioned in the methods- or 48 -as marked in Fig. 2, the exact number of trials is unclear, which is another issue), it is unclear how this task can be approached in terms of expectation creation. In impression research, expectations are typically created by providing outside information about interaction partners before an actual interaction takes place.

The potential mismatch between the theory and the study design: The authors build their research question entirely on information processing and impression formation literature which mostly focuses on one-on-one interactions. Importantly, however, the study design includes interaction with (ostensibly) two individuals (which is revealed very late in the paper, even omitted in Fig. 1), which entails a group process. Indeed, the finding on the positivity bias can be explained via group processes too: the once "dishonest" adviser who becomes honest over the course of the task can be perceived as a mere obedient and a phony and thereby disliked more than the once "honest" adviser who only realizes later that they can lie for the purpose of the task. These group processes are not considered at all in the theorization of this work. I think, given the theoretical interest of the current work, the authors should have considered a between-subjects design (an interaction with an initially honest or dishonest adviser) or clarified in separate blocks that the interactions are one-on-one.

No stopping rule or power analysis: I understand that the neuroscientific data can be challenging to collect, but I worry that 35 participants may not meet the statistical power criteria for some of the main analyses (especially the mediation analysis). Researchers, unfortunately, do not report a sensitivity analysis or even a stopping rule for data collection.

Unclarity in the procedure: As far as I understand, the participants complete the take advice game in the scanner. How does that work exactly? What is the exact setup for the task, and how do participants respond, etc.? I think the readers would benefit from these key details about the procedure. Also, what is the exact purpose of the third block where advisers were behaving similarly? Finally, the methods section says participants completed four runs; what do "runs" refer to here?

Incomplete or unclear analyses: The theoretical foundation of some of the reported analyses is missing, which worries me about potential cherry-picking. For instance, the relationship between loneliness and negative expectation - which I believe refers to negative impressions after the first block, regardless of the adviser's level of honesty. Why did the authors decide to run this analysis instead of examining the relationship between loneliness and negative first impressions for (initially) honest and (initially) dishonest individuals separately? Also, I did not see any previous work (theoretical or empirical) mentioned regarding the surprise signals analyzed in the current paper. Also, the authors mentioned that reduced OFC among lonely individuals likely leads to information of new information in a biased manner (p. 15). But isn't that something they can test with their data (e.g., whether reduced OFC among lonely individuals predicts a lower tendency to listen to the adviser who was previously honest)? Finally, although I found the OFC-TPJ coupling analyses interesting, it was unclear to me why the authors did not report the effects as a function of loneliness. Does such coupling predict trustworthiness regardless of the level of loneliness? As this effect is interpreted as a potential buffer for the negative outcomes of loneliness, it seems critical to run that analysis.

Minor Issues

- I noticed typos throughout the manuscript:
"brain regions that plays a role" p. 4
- The colors of the Fig2.e seem a bit misleading (as the analyses do not seem to include the honesty of the adviser)
- Instead of saying honest/dishonest adviser throughout, it may help to clarify that the terms are about the patterns in the first block only (maybe something like "initially honest adviser").

Response to Reviewers' Comments

Reviewer #1:

The current study provides very interesting insights into impression formation in lonely individuals regarding trust and honesty. The results indicate that lonely individuals show stronger negative expectations about others and a biased information updating for negative first impressions. Moreover, this negativity bias in information updating was accompanied by reduced orbitofrontal cortex activity. The study addresses a very important and insufficiently studied topic and I highly appreciate the authors approach of combining behavioral, computational, and fMRI methods. The manuscript is well-written and the study is well-designed. I only have some minor remarks which might help to improve the manuscript and data presentation:

We thank the reviewer for this positive evaluation of our work.

- The authors introduce cognitive biases related to loneliness in a comprehensible manner. Nevertheless, various studies investigated the relationship of loneliness with interpersonal trust in particular (e.g., Rotenberg et al., 2010, doi: <https://doi.org/10.1177/0146167210374957>; Lieberz et al., 2021, doi: <https://doi.org/10.1002/adv.202102076>) which should be mentioned when introducing this topic.

We thank the reviewer for pointing us to this literature. We have discussed these papers in the Introduction as follows (lines 38-41):

“Such cognitive biases have been proposed to be central to social withdrawal in lonely individuals, as they foster unfavorable expectations of others that induce lower levels of trust (Lieberz et al., 2021), more negative trust beliefs (Rotenberg et al., 2010), and avoidance of social relationships (J. T. Cacioppo & Hawkley, 2009).”

Moreover, in lines 75-77 the authors state that the neural basis of biased trust in loneliness is still unknown. However, this question was addressed by Lieberz et al. (2021) as well.

We thank the reviewer for pointing to this previous work. The focus on our work was on biased formation of impressions of others' trustworthiness. To this end, we look at differential learning patterns that might contribute to more negative impressions of others in lonely individuals. We then studied the neural implementation of these learning patterns in an interactive MRI experiment. Hence, in lines 76-78 we write:

“the brain regions that play a role in the formation of others' trustworthiness impressions and that are affected by subjective feelings of loneliness are still unknown.”

Hence, the focused of this paper is on learning mechanisms. On the contrary, the interesting paper by Lieberz et al. 2021 looked at the differential engagement of brain regions in lonely individuals underlying trust propensities in an economic game—the investment game. Importantly, participants in the study by Lieberz and colleagues did not learn about the trustworthiness of the trustees and were asked to entrust money with different trustees without information about these trustees and without feedback on their reciprocity in response to participants' trust decisions—equaling a classic study on trust propensity in single, anonymous interactions. On the contrary, in our study, participants interacted multiple times with the same co-player and could learn their honesty/dishonesty and adapt their behavior accordingly. We clarified this crucial difference in our manuscript in lines 78-82:

“Specifically, despite initial evidence on the brain regions underlying trust propensities during single interactions in lonely individuals (11), the neural underpinnings of social information integration during multiple one-to-one interactions and their differential recruitment and contribution to varying cognitive biases in lonely individuals are still unexplored.”

- For all reported statistical effects, effect sizes should be provided. This is especially true given the interpretation of the reported effects as “strong” (see, for example, line 124).
We thank the reviewer for mentioning this work. We now report the effect size for all of our analyses.

- Some effects are only reported in figure legends but not in the main text. This concerns OFC and TPJ activity (figure 3c/4a) but also the correlation of loneliness with OFC activity (figure 5a). I strongly suggest to report all statistical effects comprehensively in the main text as well.

We deeply apologize for this. In those figure captions, we reported the statistical threshold for multiple corrections, as we thought that someone looking at the fMRI results would want to know that information. The multiple correction thresholds were reported in detail in the Methods section. That is, for second-level analyses in lines 631-633:

“Results were whole-brain corrected for multiple comparisons using a voxel-level threshold of $p < .001$ and a family-wise error, cluster-level (FWE_c) corrected threshold of $p < .05$ ”

For functional connectivity analyses in lines 685-686:

“Significant connectivity was assessed with a voxel-level threshold of $p < .001$ and an FWE_c cluster-level threshold of $p < .05$ ”

For the whole-brain, subject-level regression (line 639):

“using $FWE_c < .05$ with a voxel-level threshold of $p < .005$ ”

Finally, we are now reporting the direct effects of the mediation analysis in the text (lines 283-286):

“Confirming our previous correlation tests, the OFC entertained significant negative relationships with loneliness (effect a: $\beta = -0.10$; $SE = 0.03$; $p < .01$) and the negativity bias (effect b: $\beta = -0.12$; $SE = 0.06$; $p < .05$) in the mediation analysis”

- Figure 4c is missing

We apologize for this. We meant Figure 4b.

- Lines 194-198: The interpretation of the relationship of brain activity with the negativity bias is interpreted in a too directional manner. This is especially true as the interpreted direction of the relationship is reversed later on.

We thank the reviewer for pointing us to this. We have revised the wording as follows (lines 208-212):

“Importantly, neural activity in the OFC and caudate was negatively associated with the impression-induced negativity bias, indicating that more negative first impressions of an adviser correlated with reduced OFC activity during the encoding of information about

that adviser (**Fig. 4a & Tab. S2**). This suggests that OFC activity during learning is central to the formation of accurate social impressions.”

- The analysis focuses on the OFC activity which is plausible given the observed association of OFC activity with the negativity bias. However, the caudate showed a significant association with the negativity bias as well and the bilateral TPJ encoded positive surprise signals. Moreover, the TPJ was introduced as being important to track belief updating. Why wasn't the TPJ activity analyzed in a more comprehensive way with regard to loneliness? I would recommend to additionally report associations of loneliness with TPJ and caudate activity. If a relationship of loneliness with brain activity is specific for OFC activity, this might further strengthen the importance of reduced OFC activity in belief updating in loneliness.

We thank the reviewer for the opportunity to clarify this point. First of all, the reviewer is right that the caudate is significantly associated with the negativity bias and that the TPJ also encoded positive surprise signals. However, our series of independent whole-brain analyses revealed convergent evidence that the OFC was the brain region central to biased information processing for both its role in information integration as revealed by our parametric model (e.g., Fig. 3c) and its relationship with the model-based negativity bias as revealed by our whole-brain regression at the second level (e.g., Fig. 4a). Moreover, the OFC was the only brain region that correlated with feelings of loneliness (e.g., Fig. 5a). Notably, this was a second, independent, whole-brain analysis run at the second level and not a follow-up ROI analysis based on our previous results. We specified this in the manuscript (lines 229-234).

“As lonely individuals judged initially dishonest advisers as less trustworthy than initially honest advisers despite all advisers having the same degree of honesty, they might have engaged the OFC less optimally during the encoding of information about those advisers. To test this, we performed a whole-brain regression analysis with loneliness scores predicting neural activity encoding honesty information about the advisers.”

Hence, these different analyses provided convergent evidence of the involvement of the OFC in biased information processing and feelings of loneliness, which is the reason why we analyzed the mediation role of this region in the relationship between loneliness and the negativity bias. We mentioned this as follows (lines 264-269):

“As these findings provide convergent evidence that negative impressions are linked to reduced activity in the OFC, which was associated with both a stronger negativity bias and greater subjective feelings of loneliness, we next set out to directly test whether the relationship between loneliness and the observed negativity bias is mediated by a less engagement of the OFC.”

Importantly, the OFC clusters resulting from these different analyses overlapped in a common area of the OFC, which was the basis of our subsequent mediation analyses (e.g., Fig. 5b). We highlighted this in the manuscript in lines 236-238:

“Importantly, the OFC cluster from this analysis overlapped with the OFC cluster that we previously observed to correlate with a stronger negativity bias”

And in lines 270-273:

“Neural activity for the mediation analysis was extracted from the OFC brain area derived from the overlap of the two OFC clusters that were independently identified in the previous two analyses on the relationships of OFC activity with the negativity bias and loneliness.”

However, we agree with the reviewer that additional analyses on the relationships of loneliness with the TPJ and caudate might help clarify the specificity of our OFC results. Hence, we set out to determine whether activity in these two brain regions were associated with participants' feelings of loneliness. To this end, we ran individual, Bayesian regression analyses to determine the individual relationships of these brain regions with loneliness, and a subsequent multivariate model with neural activity from all three brain regions to test those individual associations by controlling for the variance explained by the other predictors. These analyses revealed that the OFC had the strongest association with feelings of loneliness and was the only significant predictor in the multivariate model, strengthening the importance of reduced OFC activity in belief updating in loneliness. We reported this in the manuscript in lines 242-262:

“Finally, given the previously observed importance of the TPJ in tracking belief updating and of the caudate in the negativity bias, and due to the relevance of these brain regions in the literature on social learning and trust behaviors, we further tested with follow-up analyses their potential associations with individual feelings of loneliness and, consequently, the relative robustness of the observed relationship between OFC activity and loneliness. Bayesian models were run with neural activity from the OFC, TPJ and caudate during information encoding predicting subjective feelings of loneliness. These analyses confirmed the strong negative relationship between loneliness and the OFC ($\beta = -0.48$ (.16), 89% *high posterior density interval* (HDI) [$-0.75, -0.22$]) and revealed a weaker association between loneliness and the caudate ($\beta = -0.33$ (.18), 89% HDI [$-0.62, -0.06$]), while no significant relationship was observed between loneliness and the TPJ ($\beta = -0.02$ (.19), 89% HDI [$-0.31, 0.28$]). Importantly, when the three predictors were entered in the same multivariate model to control for the effects of the others, most of the variance explained by the caudate was explained away by the OFC, which remained the only significant predictor (OFC: $\beta = -0.49$ (.24), 89% HDI [$-0.89, -0.11$]; caudate: $\beta = -0.01$ (.25), 89% HDI [$-0.38, 0.40$]; TPJ: $\beta = -0.02$ (.17), 89% HDI [$-0.29, 0.27$]). These additional findings strengthen the importance of reduced OFC activity in belief updating for impression formation in loneliness, indicating that among the brain regions that played a role in social information encoding and the negativity bias in social learning in our study, the OFC was the one most central to subjective feelings of loneliness.”

- Take advice game: Did participants believe that the adviser was played by another participant? It is not clear to me why the adviser should lie, which could prevent the subjects from believing that it was another real person. Was there a cover story that the adviser could also increase their own outcome which might have made it necessary to lie? This is important given that the social context is mandatory to measure trust in contrast to learning in a non-social context.

We thank the reviewer for giving us the opportunity to clarify this point. Participants weren't told about the specific motivations and rationales behind the advisers'

behaviors because we wanted to avoid priming them and experimentally inducing any bias in participants' prior expectations that could have impacted their learning patterns. On the contrary, we wanted to let them learn the partner's behavior trial-by-trial and make inferences on their potential motivations based on the observed behavior. This allowed us to study how potential biases like negativity biases emerge from trial-by-trial learning and minimize any confounds that could have been induced by the instructions. Previous work has shown that instructions with information like the one mentioned by the reviewer strongly biases participants in their learning (Diaconescu et al., 2014; Mahmoodi et al., 2018; Yaniv & Kleinberger, 2000). Hence, we aimed to avoid manipulating participants priors to be able to capture a negativity bias purely due to their learning patterns and inferences on their partner's type. Indeed, previous work has shown that lonely individuals tend to have overly negative inferences in social situations with such uncertainty (Hanssen et al., 2022). We have explained this in the Methods section in lines 493-497:

"Importantly, we did not provide any information about the potential rationales behind the advisers' behaviors, as we were interested in investigating how learning biases, especially the negativity bias in lonely individuals, emerge from participants' trial-by-trial learning patterns, and we thus wanted to avoid priming and experimentally inducing any bias in participants' prior expectations that could have impacted their learning and belief updating."

To guarantee participants' credibility in the social interactions during the game, we implemented an extensive procedure before starting the experiment. Participants and their co-players were invited to the lab and instructions about the experiment were provided. So, participants saw that there were two more participants in the lab, and they were told that they were going to play with each other. They were all instructed together about the potential roles in the game and that these roles were randomly assigned. For role assignment, participants and their co-players needed to draw a ball from a lottery box before starting the experiment. They did this in separate room to avoid that the other participants knew which role they received. Further, to guarantee anonymity during the experiment, participants and their co-players needed to choose an avatar that represented themselves at the beginning of the game. In truth, when participants started the game, they all received the role of advisee. We clarified this in lines 460-466:

"Participants and their co-players were invited to the lab and instructions about the experiment were provided. They were instructed that the roles in the game were randomly assigned. For role assignment, participants and their co-players needed to draw a ball from a lottery box before starting the experiment. Further, to guarantee anonymity during the experiment, participants and their co-players needed to choose an avatar that represented themselves at the beginning of the game (Fig. S1). In truth, participants always received the role of advisee."

Moreover, we ran an exit questionnaire that collected participants' credibility in the task. In particular, we told participants that due to anonymity, we had to use avatars in the experiment and for statistical reasons, we were interested in knowing whether they believed they were playing with other participants in the task. We had two

independent raters code participants' answers. Results show that most participants believed they were playing with other participants. We reported these analyses in the Methods section.

“Finally, in an open question at the end of the experiment, participants were asked to report whether they believed they were playing with other participants during the game. To avoid social desirability effects, we told participants that consistently with the cover story at the beginning of the experiment, we had to use avatar during the game due to anonymity reasons and that now, for statistical purposes, we were interested in knowing whether they genuinely thought they were playing with the other participants during the game. Two independent raters coded participants' written responses. Results show that about 78% of our participants believed they were playing with others.”

- Please provide a more comprehensive description of the sample with regard to loneliness and general trustworthiness expectations.

We have now added those descriptions in the Methods section. For loneliness in lines 516-517:

“Loneliness scores ranged from 20 to 45 with a mean of 28.52 (median=46; SD=6.84)”

For trustworthiness expectations in lines 518-519:

“Scores ranged from 19 to 56 with a mean of 42.06 (median=43; SD=8.98).”

Reviewer #2 (Remarks to the Author):

The present research tackles several important questions regarding impression formation and updating. First, the authors investigate the presence of a negativity bias in forming impressions (weighting negative first information more than positive). Second, the authors investigate a negativity bias that is especially likely to be observed among lonelier individuals. Finally, the authors investigate a potential link between orbitofrontal cortex (OFC) activity and negativity bias, with the goal of seeing whether the OFC activity can explain the hypothesized link between loneliness and negativity bias and first impression.

To start with the strengths of the present work, I would like to applaud the researchers for raising such interesting questions and aiming to address an important gap in the literature on impression formation. Their interdisciplinary approach is valuable and innovative.

We thank the reviewer for the applause and the positive feedback on our work.

Unfortunately, however, I refrain from recommending the current work for publication. Below you can find a list of major and minor issues I observed during my review. Please note that my review excludes the details regarding the computational models as I am not an expert in that area.

We thank the reviewer for raising both major and minor issues. We have tried to address them as best as we could.

Major Issues

The issue of causality: Throughout the paper, the authors claim a causal relationship between loneliness and negativity bias. However, as the present research does not include

experimental manipulation of loneliness, it is not possible to infer any directionality. Technically, it is possible that individuals who happened to weigh negative information more heavily in the advice-taking task reported feeling lonelier at the time of the experiment (as a psychological state), or there could be a bidirectional relationship between these variables. As one of the biggest premises and the theoretical foundation of the present paper is the social cognitive outcomes of loneliness, an experimental manipulation (or at least a longitudinal design) would be essential for this particular research question.

We thank the reviewer for the opportunity to clarify this point. The reviewer is right that the uncovered relationship is correlational. We apologize if we used a too causal language for the interpretation of our results and have revised the manuscript to make sure this is not the case. We agree that the relationship between the negativity bias and loneliness could go both ways and even be directional. This is definitely an interesting question for future studies. Hence, we have tried to be more careful in explaining the relationship between loneliness and the negativity bias. We at best mention that loneliness is characterized by a stronger negativity bias, leaving open the temporal dynamics between the two. The only directional relationship we do propose is that such biased weighting of negative information might over time lead to the more negative social expectations and, in general, that more negative outlook on social life that we and others have reported in lonely individuals (for example, in lines 370-374):

“Hence, these results indicate that loneliness is characterized by biased learning dynamics leading to the formation of more negative impressions and evaluations of others, which on the long run might give rise to lonely individuals’ more negative expectations of social partners.”

Moreover, we are hesitant to suggest that these temporal dynamics between loneliness and the negativity bias should be tested by experimentally manipulating loneliness. Not only because feelings of loneliness refer to a subjective psychological state that is hardly accessible with objective manipulations. Previous studies have tried to do so but have managed to tackle only social exclusion and isolation (e.g., in a recent study that isolated participants for 10 hours) (Tomova et al., 2020), leading only to small changes in social craving. But most importantly because we believe that given how disruptive of a social life and mental well-being loneliness can be (Tackling Loneliness Evidence Review, n.d.), it should be treated as any other clinical disorders like depression and anxiety (Mann et al., 2017) and investigated like any other clinical disorder by using a combination of questionnaire-based assessments, ecological, social paradigms and computational modeling for mechanistic insights like we did in our task. Eventually, in online or population-level studies that allow the collection of a bigger sample size, advanced statistical techniques like stratified sampling (Miller & Chapman, 2001) should be employed to systematically control for an array of individual and personality characteristics and test the temporal ordering between loneliness and biased information weighting. Hence, we tend to more strongly agree with the reviewer that future studies should additionally implement a longitudinal approach to gain insights into the temporal dynamics of the mechanisms underlying loneliness and the related negativity bias. We clarified this in the Discussion of the manuscript (lines 419-430):

“A potential limitation of this study is that we did not manipulate subjective feelings of loneliness. However, we believe loneliness should be treated like other clinical disorders

such as depression and anxiety (Mann et al., 2017) and hence investigated by using a combination of questionnaire-based assessments, ecological social paradigms and computational modeling to gain insights into the mechanisms put in place by lonely states, like we did in this work. Moreover, in future work especially with bigger samples, additional, advanced statistical methods like stratified sampling with a longitudinal approach (Miller & Chapman, 2001) should be employed to more closely isolate the operating dynamics of loneliness from other factors (e.g., personality, psychoses), test how the biases in learning and impression formation highlighted in the current work lead to the emergence of overly negative social expectations in lonely individuals, and study their temporal dynamics (Fett et al., 2022; Mund & Neyer, 2019)."

Conceptual unclarity: There are a few concepts that remain unclear and potentially misleading in the writing. First, it is unclear how the authors operationalize "negativity bias" throughout the text -when are they referring to computational, and when are they referring to behavioral outcomes? I found myself completely lost trying to disentangle their results.

We apologize for this confusion. In our work, negativity bias always refers to the computational modeling results. In particular, negativity bias refers to the difference between the weighting on negative and positive information about the advisers' behavior. We have clarified this in the results (lines 148-150):

"In particular, negative first impressions led to a negativity bias (i.e., the difference between the weighting on negative and positive information about the advisers' behavior)"

And in the Methods (lines 599-604):

" τ is the honesty learning parameter, δ is the dishonesty learning parameter and S_t is the social surprise signal (i.e., $I_t - V_{t-1}$, where I_t is the type of social information received on trial t). Importantly, both τ and δ were estimated separately for each of the advisers. An individual's negativity bias was operationalized as the difference between δ and τ , pointing to the imbalanced weighting between positive and negative information about the adviser"

We have further revised the whole manuscript to ensure that the use of the term is consistent across the paper and only refers to this operationalization.

Second, the authors cite a few studies on initial expectations in their introduction (p. 4). Then they refer to the first block in their take advice game as "expectations" or "first impressions" interchangeably (p. 10). As many rounds of interactions take place in the first round (66 -as mentioned in the methods- or 48 -as marked in Fig. 2, the exact number of trials is unclear, which is another issue), it is unclear how this task can be approached in terms of expectation creation. In impression research, expectations are typically created by providing outside information about interaction partners before an actual interaction takes place.

First of all, we are sorry about the confusion of the terms "expectations" and "first impressions". However, these are indeed two different concepts and refer to different dimensions in the manuscript. Expectations refer to participants' general expectations of others' trustworthiness measured based on a questionnaire. We clarified this in lines 513-515:

“To acquire data on subjective feelings of loneliness and the participants’ general trustworthiness expectations, participants filled out two questionnaires at the end of the experiment.”

And in lines 517-518:

“For general trustworthiness expectations, participants completed the preference survey module for trust preferences.”

On the contrary, first impressions refer only to the impressions participants formed during the first block of the task. For example, in the Results (lines 179-181):

“loneliness was associated with a stronger negativity bias for advisers of whom participants formed negative first impressions in the first block of the task”

In lines 120-121:

“At the beginning of the task (block 1, 48 trials), participants formed accurate first impressions of each of the advisers’ behaviors”

And in the Methods (line 484):

“In the first, impression-formation block”

Moreover, we apologize for the confusion with respect to the number of trials. 66 refers to the total number of trials per run in the MRI. We clarified this in lines 502-503:

“Participants played a total of 4 runs (i.e., fMRI scanning sequence) in the scanner with 66 trials each for a total of 264 trials.”

However, each block of interactions had 48 trials, as the reviewer noted. Participants played 24 trials with each of the advisers. In the first 24 trials, participants could form impressions of the different advisers. We clarified this in lines 120-121:

“At the beginning of the task (block 1, 48 trials), participants formed accurate first impressions of each of the advisers’ behaviors (24 trials each)”.

The reviewer is right in mentioning that in some impression research, participants need to form impressions of a partner based on information provided to them prior to the interaction with the partner (if any). This type of experimental settings is similar to some research on social priming. However, an important review on impression formation a few years ago distinguished three different experimental paradigms that have been employed to investigate impression formation (Ames et al., 2011). In particular, secondhand information (being told about someone), direct behavioral experience (interacting with someone), and appearance (seeing someone’s look). The very first study on impression formation by Asch (Asch, 1946) could be seen as being of the first type. Studies on the impact of facial information could be thought of being of the last type (Furl, 2016). Finally, our work would fit the second type of research. Importantly, given that the type of the information provided to participants in these different experimental paradigms has different degrees of informativeness and ambiguity, the number of datapoints necessary for participants to form stable impressions of their partners varies across these experimental paradigms. For instance, even though semantic information is extremely rich, Asch-like experiments provide participants with no less than 12 datapoints (Sullivan, 2019). On the contrary, to form

stable impression from interactions with others in highly controlled, social experiments, more datapoints are required. Particularly in our study, participants didn't receive direct information about the honesty and dishonesty of the advisers but had to infer it based on feedback information on the numbers on the cards, thereby receiving low informative and highly uncertain information—especially in light of the stochasticity and stationarity of the advisers' behaviors. We have clarified this in the manuscript in lines 487-493:

“Based on pilot data, we observed that, given the low informativeness of the behavioral signal from the advisers, its uncertainty, stochasticity and non-stationarity, and the fact that participants received only indirect information about the honesty of the advisers, which needed to be inferred from the correctness of the advice and not from the accuracy of their decisions, twenty-four trials were required for participants to form stable impressions of each of the advisers.”

The potential mismatch between the theory and the study design: The authors build their research question entirely on information processing and impression formation literature which mostly focuses on one-on-one interactions. Importantly, however, the study design includes interaction with (ostensibly) two individuals (which is revealed very late in the paper, even omitted in Fig. 1), which entails a group process. Indeed, the finding on the positivity bias can be explained via group processes too: the once “dishonest” adviser who becomes honest over the course of the task can be perceived as a mere obedient and a phony and thereby disliked more than the once “honest” adviser who only realizes later that they can lie for the purpose of the task. These group processes are not considered at all in the theorization of this work. I think, given the theoretical interest of the current work, the authors should have considered a between-subjects design (an interaction with an initially honest or dishonest adviser) or clarified in separate blocks that the interactions are one-on-one.

We apologize for this confusion and gladly clarify these points. Our study did employ a paradigm with one-to-one interactions. In particular, the type of task we used was a multiple multi-round game in which participants played multiple trials with a single partner. In particular, participants played with two other partners but they played with them sequentially. Hence, there was no social group involved in our paradigm. Multiple multi-round games have been extensively employed in various fields such as behavioral economics, experimental psychology, cognitive science and neuroeconomics, and have been shown to more closely capture interpersonal rather than group processes. We have clarified these points in the manuscript (lines 455-460):

“This game is a sequential decision-making game played with each of the advisers over multiple trials, allowing participants to receive feedback about each adviser's advice-giving behavior in every trial and learn about the advisers' honesty trial-by-trial. This one-to-one interaction with the advisers enabled participants to form impressions of the advisers and adapt their own subsequent trusting behavior accordingly.”

No stopping rule or power analysis: I understand that the neuroscientific data can be challenging to collect, but I worry that 35 participants may not meet the statistical power criteria for some of the main analyses (especially the mediation analysis). Researchers, unfortunately, do not report a sensitivity analysis or even a stopping rule for data collection. *We are happy to clarify this point. Sample sizes vary greatly across disciplines depending on the employed experimental designs and the strength of the relative*

effects. For instance, in some neuroimaging literature (e.g., visual perception), standard sample sizes are even less than 10 participants—even as few as 5 in the following fMRI experiment (Stoll et al., 2020). In sequential decision-making paradigms with repeated decisions over hundreds of trials, like in our multi-round game entailing 264 trials per participant, common sample sizes are between 20 and 30 participants, as the effect sizes, due to the rich dataset per subject, generally range between medium and strong, and parameter estimates are generally robust due to the high number of trials leading to low within-subject variance. Specifically, the sample size for our behavioral and model-based effects was based on a previous behavioral study with the same experimental design that had a sample size of 33 participants (Bellucci & Park, 2020). The authors observed strong effect sizes for the main interaction effects in our behavioral analyses (the interaction effect between advisers and blocks on advice-taking behavior and the interaction effect between advisers' first impressions and information type—that is, positive/negative—on model parameters), reporting $\eta_p^2 = 0.12$ (behavioral effect) and $\eta_p^2 = 0.21$ (model-based effect). With such effect sizes, a sample size of just 9 and 6 participants would have been enough to reach a power of .80. We replicated those effect sizes in our sample and post-hoc power computations reveal that we reached an effect size $f = 0.30$ with an achieved power $> .99$ for both interaction effects on behavior and model-based information weighting. We report this in the Methods section (lines 466-452):

“Sample size was based on a previous behavioral study with a sample of 33 participants in which the interaction effect between advisers and blocks on advice-taking behavior had a $\eta_p^2 = 0.12$, achieving a power $> .99$, while the interaction effect between advisers' first impressions and information type (positive/negative) on model parameters had a $\eta_p^2 = 0.21$ with a power = 1 (Bellucci & Park, 2020). Similarly, in our study, we reached an effect size $f = 0.30$ with an achieved power $> .99$ for both interaction effects on behavior and model-based information weighting.”

Further, we now report effect sizes for all other behavioral and model-based effects in the manuscript. As can be seen, all effect sizes range from medium to strong and the average effect size of our behavioral effects has an achieved power = .78 (including secondary and follow-up tests). Moreover, for model fitting in our computational analyses, we ran a series of model comparisons using stringent metrics like AIC and exceedance probability estimations (see supplementary materials). For the neuroimaging results, as the reviewers probably know, it is currently challenging to compute a priori the required sample size for neural effects. However, our study has a sample size comparable to current neuroimaging studies using similar paradigms (Garrett et al., 2016; Moneta et al., 2023). Specifically, we know a few neuroimaging study employing brain-behavior mediation analyses that have even smaller sample sizes than our study (Kappes et al., 2020; Wager et al., 2008). This is because the relationships between brain and behavior that are required to be significant before a mediation can be performed tend to be very strong. For instance, in our study, it is easy to see from the beta coefficients and standard errors of the direct effects that a required sample size for a power = .80 is about 24 participants (Gelman & Hill, n.d.). Further, our mediation hypothesis was tested using bootstrapping with 10,000 permutations (in the manuscript in lines 673-674), which is considered a more robust test than other methods for mediation analyses (Shrout & Bolger, 2002).

We employed such robust tests for all other analyses as well. For instance, in binary classification analyses like the one we ran with the OFC, it might be natural to assume that the chance level be 50% and test the significance of the classification against 50%. However, it has previously been pointed out that in many cases (due to noise and spurious correlations in the datasets), the empirical chance level is effectively higher than 50% (Allefeld et al., 2016). Hence, it is advisable to run permutation tests to reduce the chance of false positives, which is what we have done in our study (lines 655-664). Similarly, we ran a series of whole-brain, independent analyses (as opposed to ROI analyses) using stringent thresholds for correction of multiple comparisons (e.g., $FEW < .05$), as previously recommended (Eklund et al., 2016). In addition, in this revision, we followed the reviewers' suggestions and ran further test and control analyses that have further strengthen our results.

We are hence confident that our findings are not only strong but robust, as different methods and analyses provide convergent evidence supporting them (subject-level, group-level, univariate, multivariate, functional connectivity) with medium to strong effect sizes. Moreover, our work applies state-of-the-art methodology (computational modeling, machine learning algorithm, Bayesian regressions), follows important guidelines of the scientific field in which it is placed, implements an experimental design with high ecological validity (repeated, multi-round decision-making task with hundreds of trials), and features a series of control tests that, thanks also to the reviewers' suggestions, have all contributed to strengthening our results.

Unclarity in the procedure: As far as I understand, the participants complete the take advice game in the scanner. How does that work exactly? What is the exact setup for the task, and how do participants respond, etc.? I think the readers would benefit from these key details about the procedure. Also, what is the exact purpose of the third block where advisers were behaving similarly? Finally, the methods section says participants completed four runs; what do "runs" refer to here?

We apologize for these omissions in the procedures and the confusion due to the technical jargon. Yes, the reviewer is correct that participants did the task in the scanner. The experimental setup is a standard setup for an fMRI study. Particularly, participants were connected with a computer outside of the scanner whose screen they could see through a mirror placed on the MRI coil. Participants were provided with MRI-compatible button boxes that they use to give their responses during the task. Moreover, "run" is jargon for an MRI scanning sequence. Due to physical constraints related to MRI signal stability and room temperature during MRI operation, the about 40-minute-long task was divided in 4 MRI scanning sequences within which the sequence stopped and participants were allowed to have a small break (as is generally done in MRI experiments). Importantly, participants did not move from their position in the scanner during those breaks. We have clarified it in the Methods (lines 502-506): "Participants played a total of 4 runs (i.e., fMRI scanning sequence) in the scanner with 66 trials each for a total of 264 trials. Participants used a standard MRI-compatible button box to make their choices in the MRI scanner. Participants had small breaks between one run and the other during which they laid in the MRI scanner and were instructed to keep still."

Finally, we had several reasons for the third block related to the need to provide a nonstationary period with similar hidden behavioral dynamics for the two advisers to

allow participants to improve their inference accuracy and enable the computational models to achieve more stable model estimates. Importantly, the degree to which participants were biased in this period allowed us to provide an additional behavioral proof of the effects of positive and negative first impressions on participants' choices and beliefs. We clarified this in the manuscript in lines 108-109:

"This allowed us to investigate the impact of first impression on participants' behavior toward advisers with similar overall honest behavior."

And we explicitly tested this in the Results (lines 130-135):

"Finally, when at the end of the interaction (i.e., in block 3) both initially honest and dishonest advisers provide advice with the same reliability rate (so that participants should have been indifferent in choosing one over the other), we observed a strong effect of first impressions on participants' choices, such as they preferred to take advice from advisers of whom they had a positive first impression ($t_{(34)} = -3.41$; $p < .002$; Cohen's $d = .58$)."

Incomplete or unclear analyses: The theoretical foundation of some of the reported analyses is missing, which worries me about potential cherry-picking. For instance, the relationship between loneliness and negative expectation - which I believe refers to negative impressions after the first block, regardless of the adviser's level of honesty. Why did the authors decide to run this analysis instead of examining the relationship between loneliness and negative first impressions for (initially) honest and (initially) dishonest individuals separately?

We apologize for this confusion. We have now clarified our hypotheses in the Introduction. We hope it's now clearer how our analyses refer to those hypotheses. In particular, the overall negative expectations of others' trustworthiness are not the same as the negative impressions participants formed in the first block of the experiment. We have addressed this in our answer to the reviewer's previous comment. We are sorry for this confusion and we tried to make our hypotheses on those analyses clearer now as follows. First, loneliness has previously been associated with more negative social expectations of social partners and social interactions (lines 31-44). We hence tested in our study whether this also holds for expectations of others' trustworthiness. These are general social expectations about the trustworthiness of other people that can be measured prior to any interaction (see our previous answer on how we measured them). Our results confirmed that participants who were lonelier were more likely to report that others are not to be trusted and hence judged as overall less trustworthy (lines 178-179):

"First, our data show indeed that lonelier individuals report to have more negative expectations of others' trustworthiness ($r(31) = -.35$; $p = .045$; Fig. 3a)."

In line with previous evidence, we hypothesized that the reason why lonely individuals might have more negative expectations of others might be linked to the fact that they might form more negative impressions of their social partners during social interactions (lines 42-44):

"Recently, it has been proposed that such biased social impressions impair social behaviors and feedback learning in social interactions, leading lonely individuals to form more negative expectations of social partners (Bellucci, 2020)."

In particular, biased social impressions might emerge due to biased integration of information about others (lines 31-41):

“In particular, feelings of loneliness have been hypothesized to negatively impact the formation of positive impressions of social partners (Wong et al., 2016). A model of how loneliness induces cognitive biases in social evaluations hypothesizes that feelings of loneliness are associated with hypervigilance for negative social cues, jeopardizing one’s social abilities (Qualter et al., 2015) and contributing to the development of depression (S. Cacioppo et al., 2015; Rico-Uribe et al., 2018). Previous evidence indicates attentional, evaluative and memory biases that lead lonely individuals to pay more attention to threatening events, give more evaluative weight to negative interactions and be more likely to remember negative interactions (Duck et al., 1994; Gardner et al., 2005; Pickett & Gardner, 2005). Such cognitive biases have been proposed to be central to social withdrawal in lonely individuals, as they foster unfavorable expectations of others that induce lower levels of trust (Lieberz et al., 2021), more negative trust beliefs (Rotenberg et al., 2010), and avoidance of social relationships (J. T. Cacioppo & Hawkley, 2009).”

And we have more clearly clarified our reasoning in lines 172-177:

“Second, we tested whether lonely individuals have more negative expectations of their partners and whether they manifest an even stronger negativity bias induced by negative first impressions of their partner’s trustworthiness. If lonelier individuals have more negative expectations of others, greater feelings of loneliness should favor the weighting of negative information especially about those of whom participants had formed more negative first impressions.”

Hence, negative impressions of initially dishonest advisers should have fostered a stronger negativity bias in lonelier individuals—a test we reported in lines 180-182:

“Loneliness was associated with a stronger negativity bias for advisers of whom participants formed negative first impressions ($r_{(31)} = .43$; $p = .012$; Fig. 3b) but not positive ones ($r_{(31)} = -.15$; $p = .392$).”

Finally, we tested whether this negativity bias depended on a stronger weighting of negative information about the initially dishonest advisers (lines 182-184).

“higher levels of loneliness correlated with a stronger weighting on negative information about initially dishonest advisers than on the same information about initially honest ones ($r_{(31)} = .47$; $p = .006$).”

Also, I did not see any previous work (theoretical or empirical) mentioned regarding the surprise signals analyzed in the current paper.

Surprise signals are a metrics of belief updating based on the divergence between expected and realized outcomes. In particular, in our study, we computed those signals based on our computational models. In the Introduction we mentioned studies linking the TPJ and OFC to model-based surprise signals using similar computational machinery. We have mentioned this evidence in lines 70-73:

“Previous work has highlighted the important role of the orbitofrontal cortex (OFC) and temporoparietal junction (TPJ) in social beliefs and social learning (Behrens et al., 2008; Bellucci et al., 2019). In particular, the TPJ has been shown to track belief updating about

others (Behrens et al., 2008; Diaconescu et al., 2017), while the OFC is involved in belief-consistent valuations of others (Kaplan et al., 2016; Knutson et al., 2007).”

We have further characterized this metrics in the Results in lines 198-200:

“we reasoned that failures of properly encoding positive surprise signals (a signature of better-than-expected outcomes) might hinder the encoding of more positive information about others’ behavior”

Moreover, in the Methods, we have gone into the details on how this metrics was computed in our computational model (lines 593-601):

“Random-effects Bayesian model comparison (Fig. S2) indicated that the winning model was a model with two learning rates (M7) weighting the type of social information separately, as follows:

$$V_t = V_{t-1} + \tau S_t \mathbf{1}_h + \delta S_t (1 - \mathbf{1}_h) \quad (3)$$

where V_t is the subjective value of trusting the adviser on trial t , τ is the honesty learning parameter, δ is the dishonesty learning parameter and S_t is the social surprise signal (i.e., $I_t - V_{t-1}$, where I_t is the type of social information received on trial t).”

Further, we have made explicit that when we talked about the neural signatures of social surprise in our manuscript, we are referring to neural correlates of those model-based estimates, for example in the Results section (lines 203-205):

“we first examined the neural signatures tracking updates of advisers’ honest and dishonest behaviors on a trial-by-trial basis by using model-based surprise signals (see equation 3 in Methods)”

And in the Methods section (lines 626-627):

“To examine the neural signatures of social surprise (model-based S estimates from equation 3)”

Also, the authors mentioned that reduced OFC among lonely individuals likely leads to information of new information in a biased manner (p. 15). But isn’t that something they can test with their data (e.g., whether reduced OFC among lonely individuals predicts a lower tendency to listen to the adviser who was previously honest)?

We thank the reviewer for pointing to this ambiguous passage. This is exactly what we tested in the passage, and we were drawing the conclusions of those tests. We adopted clearer formulations in lines 286-289:

“These results suggest that the less the OFC is engaged during integration of information about others’ behaviors, the stronger the relationship between the negativity bias in learning and feelings of loneliness, providing evidence on a potential neurocomputational mechanism underlying social learning in lonely individuals that might contribute to biased social impressions and expectations in loneliness.”

Finally, although I found the OFC-TPJ coupling analyses interesting, it was unclear to me why the authors did not report the effects as a function of loneliness. Does such coupling predict trustworthiness regardless of the level of loneliness? As this effect is interpreted as a potential buffer for the negative outcomes of loneliness, it seems critical to run that analysis.

We thank the reviewer for this very interesting suggestion. The reviewer is effectively asking whether loneliness moderates the relationship between the OFC-TPJ coupling and trustworthiness judgments (that is, whether the functional coupling predicted trustworthiness judgments differently as a function of individual loneliness levels or regardless of levels of loneliness). We tested this hypothesis and we found that, indeed, as the reviewer predicted, loneliness moderated the relationship between the OFC-TPJ coupling and trustworthiness judgments. In particular, this relationship was stronger for higher loneliness levels. These results, as pointed out by the reviewer, strengthen the interpretation of this functional coupling as a potential buffer for the negative outcomes of loneliness. We have reported this analysis and plotted the moderation effect in Fig. 6c (lines 306-314):

“Importantly, this relationship between stronger OFC-TPJ coupling and more positive trustworthiness judgments about initially dishonest advisers was moderated by participants’ feelings of loneliness. Specifically, a moderation analysis revealed that this relationship was stronger for higher levels of loneliness ($\beta = 0.30$ (.15), 89% HDI [0.04, 0.57]; Fig. 6c), suggesting that a stronger coupling between the TPJ and OFC was associated with more positive trustworthiness judgments in lonelier individuals. This further lends support to the hypothesis that such coupling might represent a buffering mechanism against the negative effects of loneliness on impression formation and learning.”

We further discuss these results in the Discussion in lines 410-418:

“Moreover, we observed that this relationship was moderated by individual feelings of loneliness. Specifically, for higher loneliness levels, the coupling between these two regions more strongly predicted more positive trustworthiness judgments of the initially dishonest adviser. These converging results suggest that the impairment of a more accurate revision of an individual’s initial negative impressions due to the decreased OFC activity during the encoding of new incoming information could have been buffered by a more efficient information exchange between the OFC and the TPJ (54). Hence, this functional pathway might reflect a compensatory mechanism for negativity biases in information weighting and impression formation, particularly in lonely individuals.”

Minor Issues

- I noticed typos throughout the manuscript:
“brain regions that plays a role” p. 4

We thank the reviewer for pointing this typo. We’ve revised it.

- The colors of the Fig2.e seem a bit misleading (as the analyses do not seem to include the honesty of the adviser)

We apologize for the confusion. Fig 2e shows the correlations between participants’ willingness to take the adviser’s advice and their trustworthiness ratings of the adviser. These were run separately for both the initially honest adviser (blue, top plot) and the initially dishonest adviser (orange, bottom plot). We clarified this better in the figure caption:

“Correlations between participants’ advice-taking behavior and trustworthiness judgments of the advisers. Top plot in blue shows the correlation for initially honest advisers, while the bottom plot in orange shows the correlation for initially dishonest advisers”

- Instead of saying honest/dishonest adviser throughout, it may help to clarify that the terms are about the patterns in the first block only (maybe something like “initially honest adviser”).

We thank the reviewer for this suggestion. We now substituted “honest/dishonest adviser” with “initially honest/dishonest advisers” and have clarified in the figure captions that “honest/dishonest adviser” is short for “initially honest/dishonest advisers”. For example, in Fig. 2:

“In the figure, ‘honest adviser’ is short for ‘initially honest adviser’ and ‘dishonest adviser’ for ‘initially dishonest adviser’.”

We have also clarified the terms in the manuscript as follows (lines 103-106):

“Importantly, some advisers started off with a dishonest behavior (i.e., initially dishonest advisers) while others showed to be honest at the beginning of the interaction (i.e., initially honest advisers)”

REVIEWERS' COMMENTS:

Reviewer #1 (Remarks to the Author):

I thank the authors for the revisions undertaken to the manuscript, which sufficiently clarified all my concerns. The additional analyses and explanations strengthen the original findings and I no longer have any concerns about publication.

Reviewer #2 (Remarks to the Author):

I now carefully read the revised manuscript of "Neurocomputational mechanisms of biased impression formation in lonely individuals," as well as the authors' responses to my previous comments. It is a pleasure to see that the authors responded to all my points and made proper revisions to the manuscript. I think the manuscript has largely improved as a result. I am especially glad to see that the moderation analysis, which was missing in the initial submission, is now added, and it confirms the authors' buffering hypothesis for OFC-TPJ coupling effects on updating impressions for initially dishonest individuals. I thank the authors for their revisions and do not have any additional comments.